# Structural basis for receptor selectivity and inverse agonism in S1P$_5$ receptors

Elizaveta Lyapina [1,18], Egor Marin [1,2,18], Anastasiia Gusach [1,15,18], Philipp Orekhov [1,3,4], Andrey Gerasimov [5], Aleksandra Luginina [1], Daniil Vakhrameev [1], Margarita Ergasheva [1], Margarita Kovaleva [1], Georgii Khusainov [1,16], Polina Khorn [1], Mikhail Shevtsov [1], Kirill Kovalev [1,6], Sergey Bukhdruker [1], Ivan Okhrimenko [1], Petr Popov [1,17], Hao Hu [7], Uwe Weierstall [7], Wei Liu [8], Yunje Cho [9], Ivan Gushchin [1], Andrey Rogachev [1,10], Gleb Bourenkov [6], Sehan Park [11], Gisu Park [11], Hyo Jung Hyun [11], Jaehyun Park [11,12], Valentin Gordeliy [13], Valentin Borshchevskiy [1,10] ✉, Alexey Mishin [1] ✉ & Vadim Cherezov [14] ✉

The bioactive lysophospholipid sphingosine-1-phosphate (S1P) acts via five different subtypes of S1P receptors (S1PRs) - S1P$_{1-5}$. S1P$_5$ is predominantly expressed in nervous and immune systems, regulating the egress of natural killer cells from lymph nodes and playing a role in immune and neurodegenerative disorders, as well as carcinogenesis. Several S1PR therapeutic drugs have been developed to treat these diseases; however, they lack receptor subtype selectivity, which leads to side effects. In this article, we describe a 2.2 Å resolution room temperature crystal structure of the human S1P$_5$ receptor in complex with a selective inverse agonist determined by serial femtosecond crystallography (SFX) at the Pohang Accelerator Laboratory X-Ray Free Electron Laser (PAL-XFEL) and analyze its structure-activity relationship data. The structure demonstrates a unique ligand-binding mode, involving an allosteric sub-pocket, which clarifies the receptor subtype selectivity and provides a template for structure-based drug design. Together with previously published S1PR structures in complex with antagonists and agonists, our structure with S1P$_5$-inverse agonist sheds light on the activation mechanism and reveals structural determinants of the inverse agonism in the S1PR family.

[1]Research Center for Molecular Mechanisms of Aging and Age-related Diseases, Moscow Institute of Physics and Technology, Dolgoprudny 141701, Russia. [2]Groningen Biomolecular Sciences and Biotechnology Institute, University of Groningen, Nijenborgh 4, 9747 AG Groningen, The Netherlands. [3]Faculty of Biology, Lomonosov Moscow State University, Moscow 119991, Russia. [4]Faculty of Biology, Shenzhen MSU-BIT University, Shenzhen 518172, China. [5]Vyatka State University, Kirov 610020, Russia. [6]European Molecular Biology Laboratory, Hamburg unit c/o DESY, Hamburg, Germany. [7]Department of Physics, Arizona State University, Tempe, AZ 85281, USA. [8]Cancer Center and Department of Pharmacology and Toxicology, Medical College of Wisconsin, Milwaukee, WI 53226, USA. [9]Department of Life Science, Pohang University of Science and Technology, Pohang, Republic of Korea. [10]Joint Institute for Nuclear Research, Dubna 141980, Russia. [11]Pohang Accelerator Laboratory, POSTECH, Pohang 37673, Republic of Korea. [12]Department of Chemical Engineering, POSTECH, Pohang 37673, Republic of Korea. [13]Institut de Biologie Structurale (IBS), Université Grenoble Alpes, CEA, CNRS, Grenoble 38400, France. [14]Bridge Institute, Department of Chemistry, University of Southern California, Los Angeles, CA 90089, USA. [15]Present address: MRC Laboratory of Molecular Biology, Cambridge CB2 0QH, UK. [16]Present address: Division of Biology and Chemistry, Paul Scherrer Institute, Forschungsstrasse 111, 5232 Villigen, PSI, Switzerland. [17]Present address: iMolecule, Skolkovo Institute of Science and Technology, Bolshoy Boulevard 30, bld. 1, Moscow 121205, Russia. [18]These authors contributed equally: Elizaveta Lyapina, Egor Marin, Anastasiia Gusach. ✉e-mail: borshchevskiy.vi@phystech.edu; mishinalexey@phystech.edu; cherezov@usc.edu

Sphingosine-1-phosphate (S1P) is a lysosphingolipid bio-regulator produced from ceramide in activated platelets, injured cells, and cells stimulated by protein growth factors[1,2]. S1P is released in the blood[3], where it regulates angiogenesis[4], cell proliferation, migration, and mitosis[5] by activating five subtypes of the S1P G-protein-coupled receptors−S1P$_{1-5}$. S1P$_1$ couples only to G$_i$ protein, S1P$_4$, and S1P$_5$ signal through G$_i$ and G$_{12/13}$[6], and both S1P$_2$ and S1P$_3$ couple to G$_i$, G$_{12/13}$, and G$_q$[7]. S1P receptors (S1PRs) have different expression profiles−S1P$_1$−S1P$_3$ is expressed in all organs throughout the body, while S1P$_4$ expression is limited to the immune system, and S1P$_5$ is predominantly expressed in the nervous (oligodendrocytes) and immune (NK cells) systems[8]. S1P$_5$ also inhibits PAR-1-mediated platelet activation[9]. This receptor plays an important role in autoimmune[10] and neurodegenerative disorders[10,11] as well as carcinogenesis[12]. For example, S1P$_5$ agonists elicit neuroprotective effects in Alzheimer's and Huntington's diseases[10], while S1P$_5$ inhibition leads to apoptosis of cancerous NK cells in large granular

leukemia (LGL)[12]. Non-selective modulators such as fingolimod[13], as well as dual S1P$_1$/S1P$_5$ ligands siponimod[14] and ozanimod[15,16], have been approved for the treatment of multiple sclerosis[17], Crohn's disease[18], and other autoimmune disorders. However, the exact pharmacological role of S1P$_5$ remains unclear, mostly due to the lack of well-characterized potent and highly selective S1P$_5$ ligands with in vivo activity. While inhibition of S1P$_5$ is considered a prospective treatment for LGL[12], antagonism of S1P$_1$ leads to serious adverse effects such as lung capillary leakage, renal reperfusion injury, and cancer angiogenesis[19]. Therefore, high-resolution structures of S1P$_5$ in complex with highly selective ligands would shed light on receptor selectivity and provide templates for structure-based design of selective therapeutic drugs with more focused function and fewer side effects.

The first crystal structure of an S1PR was published in 2012[20], revealing the inactive state conformation of the human S1P$_1$ in complex with a selective antagonist sphingolipid mimetic ML056.

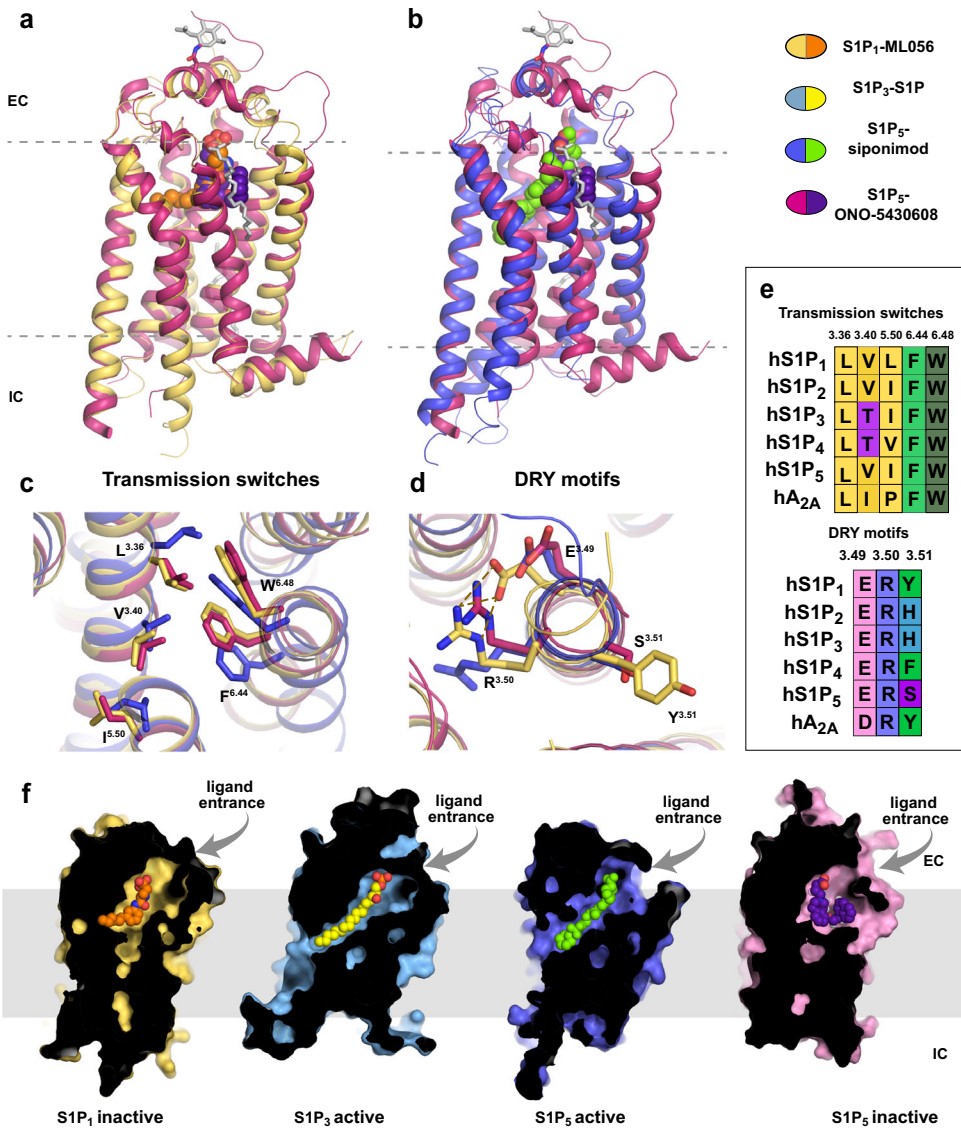

**Fig. 1 | Structure of S1P$_5$ and its comparison with structures of other S1PRs: overview and conservative motifs. a** Superposition of the obtained in this work inactive S1P$_5$ structure (pink cartoon) in complex with ONO-5430608 (purple spheres) with the inactive S1P$_1$ (yellow cartoon)-ML056 (orange spheres) complex (PDB ID 3V2Y). **b** Superposition of the inactive S1P$_5$-ONO-5430608 with the active S1P$_5$ (violet cartoon)-siponimod (green spheres) complex (PDB ID 7EW1). Glycosylated residues and lipids observed in the S1P$_5$-ONO-5430608 structure are shown as gray sticks. **c** Superposition of transmission switches for S1P$_5$-ONO-5430608 (inactive state), S1P$_5$-siponimod (active state), and S1P$_1$-ML056 (inactive state). **d** Superposition of the DRY functional motif for the same three receptor-ligand pairs as in **c**. **e** Sequence conservation of the transmission switches and DRY motif in the S1PR family. Adenosine A$_{2A}$ receptor is included as a representative receptor of class A GPCR. **f** Sliced surface representation of known structures from the S1PR family with corresponding ligands: S1P$_1$-ML056 (PDB ID 3V2Y), S1P$_3$-S1P (PDB ID 7C4S), S1P$_5$-siponimod (PDB ID 7EW1), and S1P$_5$-ONO-5430608 (this work, PDB ID 7YXA).

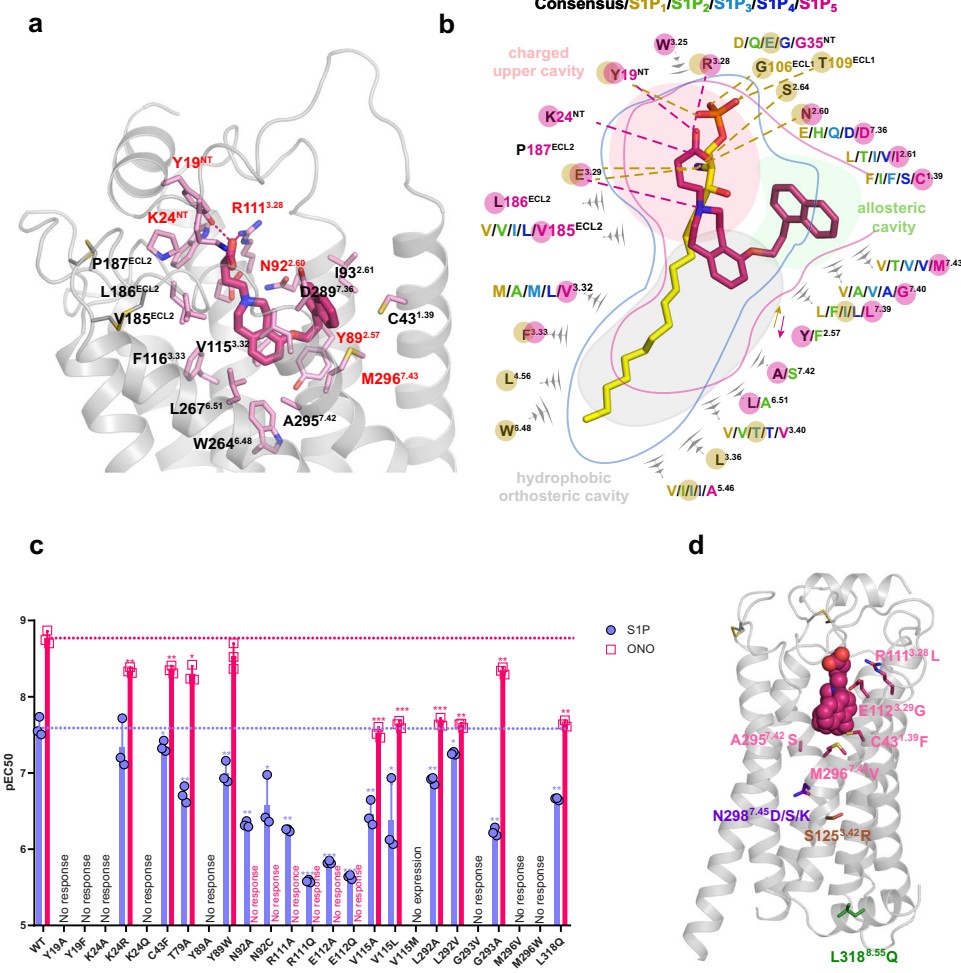

**Fig. 2 | Structural and functional comparison between ONO-5430608 and S1P binding. a** Binding pose of ONO-5430608 (pink, thick sticks) in S1P5 and its interactions with the receptor residues (light pink, thin sticks). Polar interactions are shown as dashed lines. Residues that had mutations disrupting response to ONO-5430608 are labeled in red. **b** Schematic diagram of the ligand binding pocket and interactions between ONO-5430608 and S1P5 (this work, PDB ID 7YXA) compared to interactions between S1P and S1P3 (PDB ID 7C4S). Residues are color-coded according to different S1P receptor subtypes (S1P1–yellow, S1P2–light green, S1P3–light blue, S1P4–dark blue, S1P5–pink). Black stands for the consensus residue shared by all receptors. Residues interacting with ONO-5430608 are highlighted with pink circles, residues interacting with S1P are highlighted with yellow circles.

Polar interactions are shown as dashed lines. **c** Potencies (pEC50) of S1P (purple, agonism) and ONO-5430608 (pink, inverse agonism) at WT and mutants of S1P5 in Gi protein-mediated signaling assays. Data are shown as mean ± SD of $n = 3$ independent experiments conducted in triplicates. Data were analyzed by one-sided two-sample $t$ test; $*10^{-2} \leq p < 5 \times 10^{-2}$, $**10^{-3} \leq p < 10^{-2}$, $***p < 10^{-3}$. Source data including $p$ values are provided as a Source Data file. Corresponding dose–response curves are shown in Supplementary Fig. 6. **d** Examples of naturally occurring missense SNVs in S1P5, mapped on the receptor structure. SNVs in the ligand binding pocket is shown in pink, in the sodium site–in purple, disrupting conserved hydrogen bond network–in salmon, disrupting G12/13 signaling–in green.

Recently, a crystal structure of S1P3 bound to its endogenous agonist[21], as well as cryo-EM structures of S1P1, S1P3, and S1P5 in complex with Gi proteins[22,23], and S1P1 in complex with Gi and β-arrestin[24], provided insights in the activation mechanism for the S1PR family. However, no structures of this family members in complex with inverse agonists have been reported to date.

In this article, we present the crystal structure of S1P5 in complex with a selective inverse agonist ONO-5430608[25] determined by serial femtosecond crystallography (SFX) and analyze it alongside structure-activity relationship data from cell-based functional assays using extensive mutagenesis, molecular docking, molecular dynamics, and AlphaFold simulations.

## Results

### Structure determination using an X-ray free-electron laser (XFEL)

Human S1P5 receptor was engineered for crystallization by fusing a thermostabilized apocytochrome b562RIL[26] into the third intracellular loop (ICL3) and adding a haemagglutinin signal peptide, FLAG-tag, and a linker on the N-terminus as well as a PreScission Protease site and decahistidine tag on the C-terminus (Supplementary Fig. 1). Crystals of S1P5 bound to an inverse agonist ONO-5430608 were obtained by lipidic cubic phase (LCP) crystallization[27] reaching a maximum size of 30 μm. Our initial attempts at solving the structure using synchrotron data were unsuccessful. Crystals of S1P5 bound to ONO-5430608 were then optimized to grow at a high crystal density with an average size of ~5–10 μm and used for room temperature SFX data collection at PAL-XFEL (Supplementary Fig. 2). The crystal structure was solved at a 2.2 Å resolution in the P2₁2₁2₁ space group (Supplementary Table 1). A high systematic background scattering from the direct XFEL beam (Supplementary Fig. 3) combined with pseudotranslation led to high structure refinement R-factors, although it did not affect the excellent quality of electron density maps (see Methods and Supplementary Fig. 4). The receptor crystallized with two monomers per asymmetric unit, forming an antiparallel dimer through the TM4-TM4 interface (Supplementary Fig. 5).

## Inactive conformation of S1P$_5$ in complex with ONO-5430608

The S1P$_5$ structure in complex with ONO-5430608 shares the classical architecture with other class A α-branch lipid receptors[20,21,28], including a heptahelical transmembrane bundle (7TM), two pairs of disulfide bonds stabilizing extracellular loops 2 and 3 (ECL2 and ECL3), an amphipathic C-terminal helix 8 running parallel to the membrane on the intracellular side, and an N-terminal helix capping the ligand-binding site. As expected, the receptor is captured in the inactive conformation (Fig. 1a, b) based on its overall alignment with the inactive state S1P$_1$ (PDB ID 3V2Y, Cα RMSD = 0.84/0.78 Å on 90% of residues for chains A/B of our S1P$_5$ structure) and the active state S1P$_5$ (PDB ID 7EW1, Cα RMSD = 1.40/1.40 Å on 90% of residues for chains A/B of our S1P$_5$ structure) as well as on the conformation of conserved activation-related motifs described below.

A dual toggle switch L(F)$^{3.36}$-W$^{6.48}$ (superscripts refer to the Ballesteros-Weinstein[29] residue numbering scheme in class A GPCRs) together with P$^{5.50}$-I$^{3.40}$-F$^{6.44}$ motif have been characterized as the common microswitches in class A GPCRs that transmit activation-related conformational changes from the ligand-binding pocket towards an outward movement of TMs 5 and 6 and inward displacement of TM7 on the intracellular side[30,31]. In all S1PRs, the dual toggle switch is conserved as L$^{3.36}$-W$^{6.48}$; however, the P-I-F motif deviates from the consensus, and in S1P$_5$, it is represented as I$^{5.50}$-V$^{3.40}$-F$^{6.44}$ (Fig. 1e). Nevertheless, the I-V-F motif in S1P$_5$ apparently serves a similar role as the classical P-I-F motif in other receptors, as the side chains of V$^{3.40}$ and F$^{6.44}$ switch over upon activation. The I-V-F switch in S1P$_5$ is connected to the dual toggle switch through steric interactions between F$^{6.44}$ and W$^{6.48}$, and the shift of W264$^{6.48}$ is accompanied by a rotamer switch of L119$^{3.36}$ (Fig. 1c). Similar dual (also known as "twin") toggle switch L(F)$^{3.36}$-W$^{6.48}$ has been shown to play a key role in the activation of several other receptors, such as CB1[32,33], AT1[34], and MC4[35].

An allosteric sodium-binding site located in the middle of the 7TM bundle near D$^{2.50}$ is highly conserved in class A GPCRs[36]. Binding of a Na$^+$ ion along with several water molecules in this site stabilizes the inactive receptor conformation. Upon receptor activation, the pocket collapses, likely expelling Na$^+$ into the cytoplasm[36,37]. Despite a relatively high resolution and conservation of critical sodium-binding residues, such as D82$^{2.50}$, S122$^{3.39}$, and N298$^{7.45}$, we could not locate a Na$^+$ in the electron density of S1P$_5$, most likely because of a low sodium concentration in the final crystallization buffer (~20 mM). Other residues lining the Na$^+$-binding pocket (N$^{1.50}$, S$^{3.39}$, N$^{7.45}$, S$^{7.46}$, N$^{7.49}$, Y$^{7.53}$) are also conserved in S1P$_5$, with the exception of two polar residues, T79$^{2.47}$ and S81$^{2.49}$, in the side part of the pocket, which are typically represented by two hydrophobic alanines[36].

On the intracellular side of the receptor, conserved residues E132$^{3.49}$ and R133$^{3.50}$ of the D[E]RY motif form an ionic lock that stabilizes the inactive state (Fig. 1d). Upon receptor activation, this ionic lock breaks apart releasing R133$^{3.50}$ for interaction with a G protein[38,39]. Interestingly, S1P$_5$ possesses S134$^{3.51}$ in this motif, which is seen in only 6 class A receptors out of 714, compared to a more common residue Y that is present in 66% of class A receptors.

## Overall structure of the ligand-binding pocket

The co-crystallized ligand ONO-5430608 (4-{6-[2-(1-naphthyl)ethoxy]-1,2,4,5-tetrahydro-3H-3-benzazepin-3-yl} butanoic acid) has been developed within a series of S1P$_5$-selective modulators[25] and characterized as a potent inverse agonist at S1P$_5$ in G$_i$-protein-mediated cAMP accumulation assay (EC$_{50}$ = 1.7 nM) (Supplementary Fig. 6 and Supplementary Table 2). The ligand was modeled in a strong electron density observed inside the ligand-binding pockets of both receptor molecules in the obtained crystal structure (Fig. 2a and Supplementary Fig. 4). The overall architecture of the pocket, shared by other members of the S1PR family, reflects both zwitterionic and amphipathic properties of the endogenous S1P ligand[20,21]. The pocket is occluded on the extracellular side by the N-terminal α-helix packed along ECL1 and

ECL2, with a small opening between TM1 and TM7 (Fig. 1f), which has been proposed to serve as the entrance gate for lipid-like ligands[20]. The orthosteric ligand-binding pocket in S1P$_5$ consists of a polar charged part, composed of residues from the N-terminal helix and extracellular tips of TM2 and TM3 that interact with the zwitterionic headgroup of S1P, as well as a hydrophobic cavity, lined up by hydrophobic and aromatic residues, which accommodates the alkyl tail of S1P (Fig. 2b). The negatively charged butanoic acid group of ONO-5430608 occupies the polar part of the pocket mimicking the phosphate group of S1P, the core tetrahydrobenzazepine rings fill in space in the middle of the pocket, while the naphthyl-ethoxy group unexpectedly swings over and extends into a previously unidentified allosteric sub-pocket. The subpocket is surrounded by non-conserved residues from TM1, TM2, and TM7 and opened in our structure due to a rotameric switch of Y89$^{2.57}$ compared to structures of other S1P receptors (Fig. 2a, b). The distinct amino acid composition of this allosteric subpocket suggests that it can serve as a selectivity determinant for S1P$_5$-specific ligands and makes the hallmark of the structure described in this work.

## Functional characterization of the ligand binding hotspots in S1P$_5$

To validate the observed ligand binding pose and further expand our knowledge about the ligand selectivity and relative importance of specific residues, we tested 25 structure-inspired ligand-binding pocket mutants of S1P$_5$ by a BRET-based cAMP production assay using the endogenous agonist S1P and the co-crystallized inverse agonist ONO-5430608 (Fig. 2c, Supplementary Table 2, and Supplementary Fig. 6). In line with the binding pocket structure description given above, we consequently characterize important interactions in each part.

The polar upper part of the binding pocket is highly conserved among the whole S1PR family (Fig. 2b). It consists of residues Y19$^{N-Term}$, K24$^{N-Term}$, N92$^{2.60}$, R111$^{3.28}$, and E112$^{3.29}$ and accommodates the phosphate and primary amine groups of S1P. The receptor's potential for multiple polar interactions in this region is utilized in anchoring zwitterionic groups of synthetic ligands of S1PRs. Thus, in our S1P$_5$ structure, the carboxyl group of ONO-5430608 is stabilized by polar interactions with Y19$^{N-term}$, K24$^{N-term}$, and R111$^{3.28}$, while the protonated tertiary amine group makes a salt bridge with E112$^{3.29}$, similar to interactions of the phosphate and secondary amine groups of ML056 in S1P$_1$ structure[20]. The zwitterionic headgroup of the endogenous S1P ligand bound to S1P$_3$ is shifted towards TM1, while retaining the same interactions except for the N-terminal K27[21].

The mutations disrupting polar interactions with zwitterionic ligand head groups: Y19$^{N-Term}$A/F, K24$^{N-Term}$A/Q, N92$^{2.60}$A/C, R111$^{3.28}$A/Q, and E112$^{3.29}$A/Q either fully abolish or significantly (by an order of magnitude or more) decrease the response for both ONO-5430608 and S1P (Fig. 2c and Supplementary Table 2). Notably, some mutations have different effects on S1P and ONO-5430608. While mutations of N92$^{2.60}$, R111$^{3.28}$, and E112$^{3.29}$ completely eliminate response to the inverse agonist, they only decrease the potency for S1P. A similar effect of mutations of homologous amino acids on S1P potency was previously observed for S1P$_3$[21]. In this case, each of the three amino acids independently interacts with the amine group of S1P (see PDB ID 7C4S). On the other hand, in our S1P$_5$-ONO-5430608 structure, these three amino acids are interconnected and form a stable cluster that further interacts with the tertiary amine and the carboxyl group of ONO-5430608. Thus, mutations of any of the three amino acids in S1P$_5$ would only partially perturb S1P complex, while they would disrupt the cluster and completely eliminate the binding of ONO-5430608. Although the locations of residues, known to interact with the phosphate group of S1P from either functional or structural data, are largely conserved between S1P receptors, the effects of their mutations on S1P potency are different. Namely, mutations of N-terminal Y29/19 and K34/24 to alanine render S1P$_1$/S1P$_5$, respectively, non-responsive to

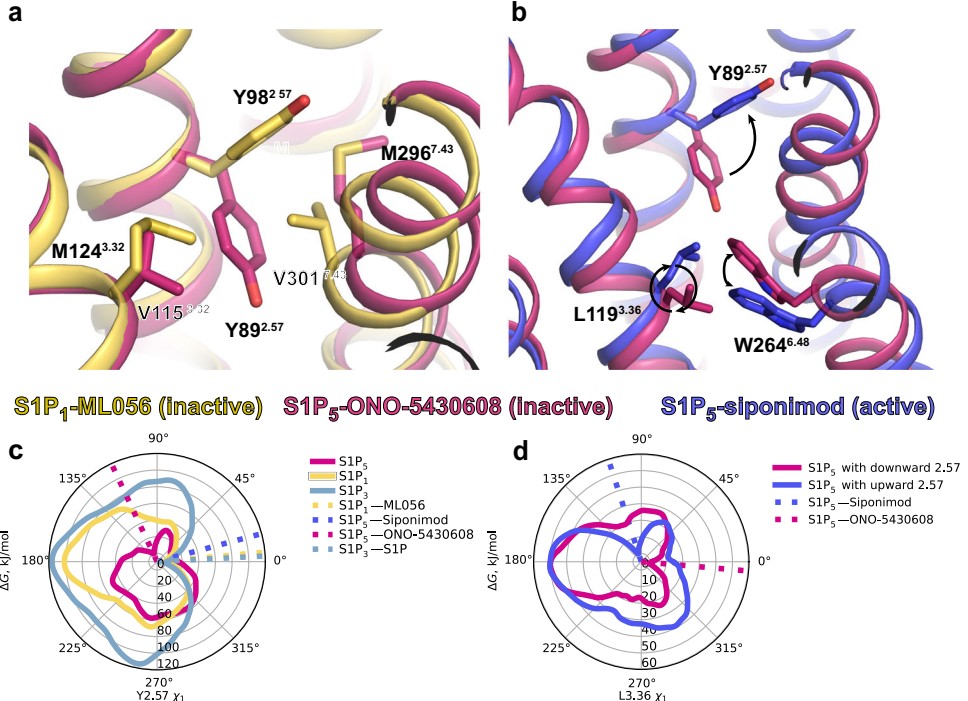

**Fig. 3 | Conformational flexibility of $Y^{2.57}$ and its effect on inverse agonism in S1P receptors. a** Two distinct upward and downward conformations of $Y^{2.57}$ as observed in crystal structures of S1P₁-ML056 (PDB ID 3V2Y) and S1P₅-ONO-5430608 (this work, PDB ID 7YXA), respectively. **b** The downward orientation of $Y^{2.57}$ is incompatible with the active state of the dual toggle switch $L^{3.36}$–$W^{6.48}$ because of a steric clash. S1P₅-ONO-5430608 (this work, PDB ID 7YXA, inactive state) is shown in pink, and S1P₅-siponimod (PDB ID 7EW1, active state) is shown in purple. **c** Free

energy profiles of the $Y^{2.57}$ side-chain torsion angle $\chi_1$ as calculated by metaMD for S1P₁, S1P₃, and S1P₅. Dotted lines correspond to $Y^{2.57}$ conformations in corresponding experimental structures. **d** Free energy profiles of the $L^{3.36}$ side-chain torsion angle in S1P₅ with two alternative orientations of $Y^{2.57}$ as calculated by metaMD. Dotted lines correspond to $L^{3.36}$ conformations in corresponding experimental structures.

S1P[20], while corresponding mutations preserve the interaction with S1P₃[21]. These data suggest a different orientation of the phosphate headgroup of S1P within the binding pocket in different receptors.

The hydrophobic part of the orthosteric binding pocket in S1PRs accommodates the lipidic tail of the endogenous ligand or its synthetic analogs such as ML056[20,21,40]. The residues on its bottom are conserved among S1PRs (Fig. 2b) and well characterized[21,40]. The top part of the hydrophobic subpocket in S1P₅, which in our structure accommodates the tetrahydrobenzazepine double-ring system of ONO-5430608, consists of residues $V115^{3.32}$, $L292^{7.39}$, and $Y89^{2.57}$ that are less characterized, although they play an important role in ligand binding. In our functional assays, mutations of $V115^{3.32}$ to A and L decrease the potencies of both S1P and ONO-5430608 (Fig. 2c and Supplementary Table 2). Additionally, $Y89^{2.57}A$ abolishes the functional response of both ligands, while $Y89^{2.57}W$ preserves it, suggesting that an aromatic residue is crucial at this position.

Although ONO-5430608 shares a similar zwitterionic headgroup with other co-crystallized S1PR ligands, its hydrophobic tail is substantially different. The bulky naphthyl group of ONO-5430608 does not fit well in the relatively narrow hydrophobic cleft of the orthosteric pocket and instead accommodates a previously uncharacterized allosteric subpocket between TM1 and TM7 (Fig. 2a, b). The subpocket is formed by non-conserved hydrophobic residues $C43^{1.39}$ (90 Å² occluded area), $I93^{2.61}$ (127 Å²), $L292^{7.39}$ (126 Å²), $G293^{7.40}$ (50 Å²), and $M296^{7.43}$ (123 Å²). Site-directed mutagenesis of residues in the allosteric pocket and functional data suggest a strong role of TM7 residues of the pocket in ligand binding. In particular, mutations $L292^{7.39}A/V$ decrease ONO-5430608 potency by over an order of magnitude, while $M296^{7.43}V/W$ and $G293^{7.40}V$ abolish the response to S1P (Fig. 2c). On the other hand, mutations $C43^{1.39}F$ and $G293^{7.40}A$ show almost no effect on ONO-5430608 potency. The strengths of the effects appear to

correlate with the occluded areas of residues interacting with ONO-5430608, as calculated from the crystal structure.

## Meta molecular dynamics simulations of $Y^{2.57}$ conformational flexibility

The allosteric subpocket displays a large variability in its residues among S1P receptors (Fig. 2b), likely contributing to the exceptional selectivity of ligands targeting it. Interestingly, this pocket is present in our S1P₅ structure largely due to the flip of one amino acid, $Y89^{2.57}$, compared to other S1PR structures. We thus explored the conformational flexibility of $Y^{2.57}$ in the available structures of S1P₁, S1P₃, and S1P₅ receptors using an enhanced molecular dynamics simulation technique, originally developed by Laio and Parrinello[41] and known as metadynamics (metaMD), as well as by targeted mutagenesis.

MetaMD facilitates sampling of the free energy landscape along the selected reaction coordinate(s), e.g., a torsion angle, by adding biasing potentials (most commonly positive Gaussians) driving the system out of local minima. By adding multiple Gaussians, the system is discouraged to return to already sampled regions of the configurational space which eventually allows it to escape free energy minima. The free energy landscape can be then recovered as the opposite of the cumulative biasing potential. Here, we used metaMD to estimate free energy profiles along the reaction coordinate corresponding to the torsion rotation of the $Y^{2.57}$ side chain.

The free energy profile of the $Y89^{2.57}$ side chain torsion in S1P₅ features two minima (Fig. 3a, c): the global minimum corresponds to a downward orientation of $Y89^{2.57}$ as observed in our crystal structure, while the second minimum at a higher energy level is close to an upward orientation of $Y^{2.57}$ found in the S1P₁ and S1P₃ crystal structures. On the other hand, the free energy profile of the $Y^{2.57}$ side chain torsion in both S1P₁ and S1P₃ has only a single minimum near their

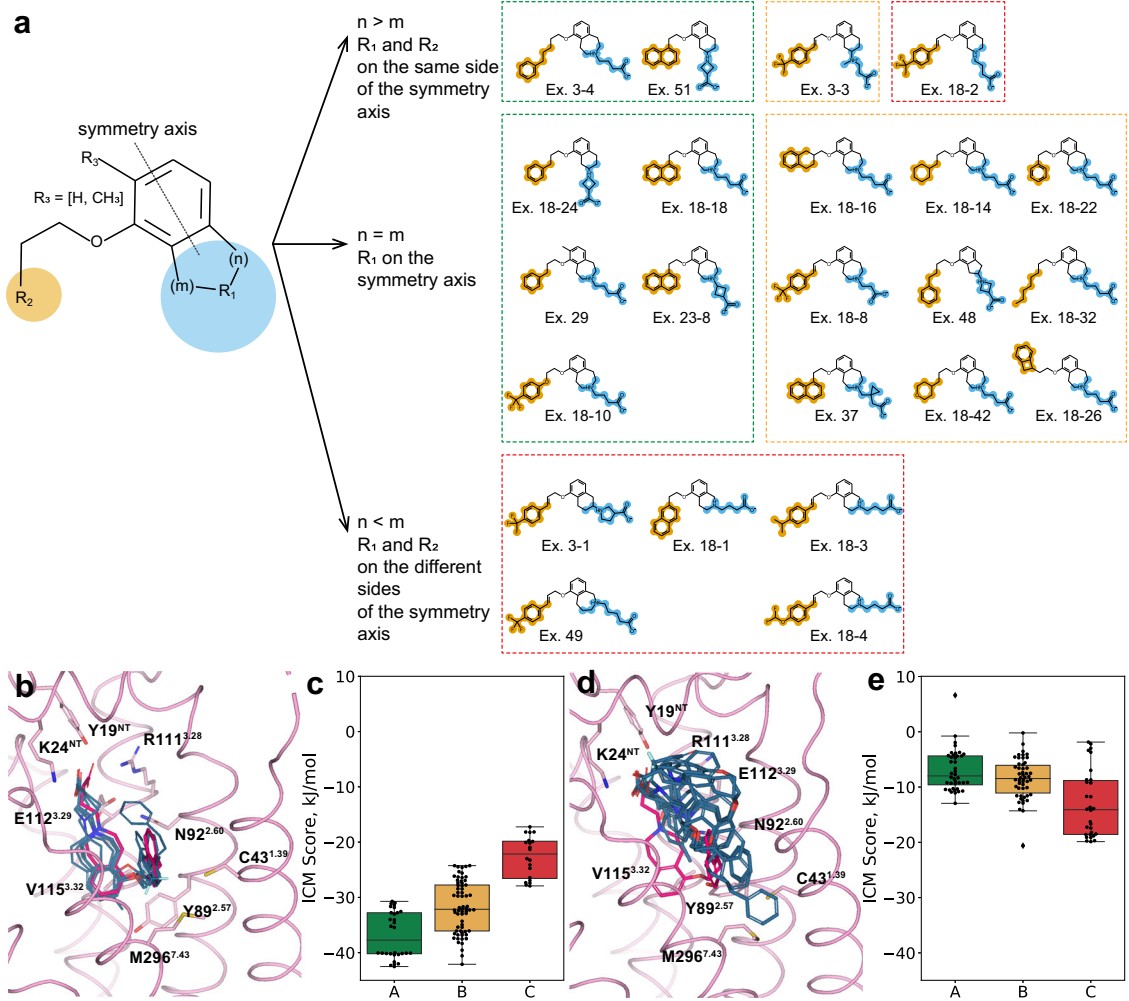

**Fig. 4 | Ligand docking simulations and substituent decomposition analysis of ONO-5430608 ligand series. a** The main scaffold and substituents. The double-ring system symmetry axis is shown as a dotted line. Rows represent different substituent's placements around the double-ring system symmetry axis: $R_1$ and $R_2$ are on the same side (top row), $R_1$ is on the axis (middle row), $R_1$ and $R_2$ are on the different sides (bottom row). Ligand groups are outlined with respect to their affinities: group 'A' (1 nM < IC$_{50}$ < 100 nM, 7 ligands) in green, group 'B' (100 nM < IC$_{50}$ < 1 µM, 10 ligands) in yellow, and group 'C' (1 µM < IC50 < 3 µM, 6 ligands) in red. **b, d** Overlay of ligand binding poses (one highest-score pose per ligand) for all group 'A' ligands docked in the S1P$_5$ crystal structure (downward conformation of Y89$^{2.57}$) (**b**) or in a metaMD snapshot with an upward conformation of Y89$^{2.57}$ (**d**). **c, e** Clustering of docking scores for all tested ligands (5 trials per ligand) corresponding to docking runs described in **b, d** respectively. All ligands are grouped according to their S1P$_5$ affinity as described in **a**. The boxplots represent the median, interquartile ranges, and whiskers within 1.5 times the interquartile range.

crystallographic upward conformations (Fig. 3c). The downward orientation of Y$^{2.57}$ in the latter case is likely hampered by steric clashes with M$^{3.32}$/V$^{7.43}$, making this conformation energetically unfavorable. S1P$_5$ has a smaller valine in position 3.32, which does not interfere with the downward orientation of Y$^{2.57}$. At the same time, a more flexible methionine in position 7.43 may swap positions with Y89$^{2.57}$ allowing the latter to switch between the upward and downward conformations.

**Insights from molecular docking**

To assess the importance of the Y89$^{2.57}$ conformation in ligand binding, we performed molecular docking of ONO-5430608 ligand series[25] (Fig. 4a) into two S1P$_5$ models: the crystal structure (Y89$^{2.57}$ in the downward conformation) and a metaMD snapshot (Y89$^{2.57}$ in the upward conformation). As expected, the docking scores correlate well with the ligand affinity[25] only in case of the crystal structure (Fig. 4b, c): the most potent group 'A' ligands (IC$_{50}$ between 1 and 100 nM) have docking scores of −37 ± 5 kJ/mol, whereas the least potent group 'C' ligands (IC$_{50}$ between 1 and 3 µm) have scores −23 ± 4 kJ/mol, and for the intermediate group 'B' (IC$_{50}$ between 100 nM and 1 µM) scores are

−32 ± 5 kJ/mol. For the metaMD snapshot, scores show no correlation with the ligand affinity (Fig. 4d, e). Accordingly, ligand docking poses also confirm that Y89$^{2.57}$ needs to be in a downward conformation for the ONO-5430608-like compounds to adopt conformations similar between each other. Namely, all of group 'A' compounds closely resemble the co-crystallized ligand pose (Fig. 4b, c). They retain interactions of the negatively-charged headgroup with Y19/K24, as well as the interaction of the positively charged amino group with E112$^{3.29}$, and the position of the double-ring system is preserved. For the upward confirmation of Y89$^{2.57}$, the docking of the group "A" ligands show no consistency between each other and the obtained data from the mutation screening (Fig. 4d, e).

Notably, the SAR data for the ONO-5430608 ligand series (Fig. 4a) suggest a role of the substituent position on the core double-ring system in the ligand binding. Namely, most of the lower affinity ligands (group "C") have a tetrahydroisoquinoline or tetrahydronaphthalene scaffold instead of the tetrahydrobenzazepine, which is more common among group "A" and "B" ligands. Likely, the affinity drop occurs due to the overall ligand shape, rather than the ring size. Namely, most ligands with both substituents placed on the same side of the middle

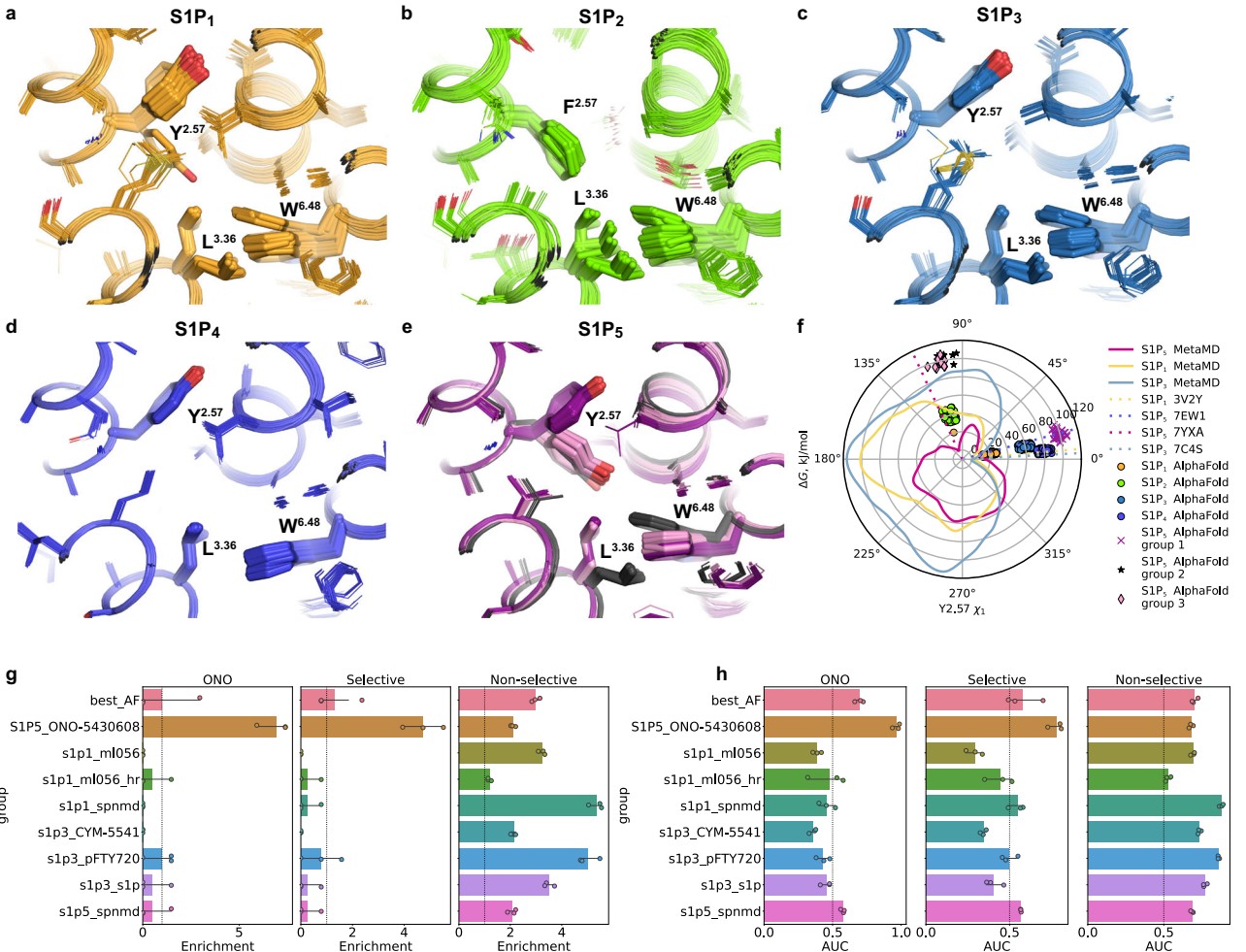

**Fig. 5 | AlphaFold prediction of S1PR structures. a–e** Conformations of $Y^{2.57}$ and the dual toggle switch $L^{3.36}$-$W^{6.48}$ in 50 AlphaFold models of each $S1P_{1–5}$ subtype, respectively. Three distinct conformations of $S1P_5$ are shown in dark violet, light violet, and black. **f** Free energy profiles of the $Y^{2.57}$ side-chain torsion angle $\chi_1$ in $S1P_1$ (yellow line), $S1P_3$ (blue line) and $S1P_5$ (pink line) calculated by metaMD. Dotted lines correspond to $Y^{2.57}$ conformations in experimental structures. Individual points correspond to $Y^{2.57}$ conformations in AlphaFold models. **g, h** Performance of existing experimental structures versus the best AlphaFold structure in virtual screening of compounds from three benchmark tests (ONO, selective, and non-selective), as judged by their AUC and enrichment (top 10%), respectively. Bar heights represent mean ± 95% CI for $n = 3$ independent docking trials with effort = 1.

plane across the double-ring system display higher affinity, while all ligands with two substituents placed on different sides have a low affinity (Fig. 4a), with the only exception, Example 18-2, which however has an amine group placed within the isoquinoline system, compared to other same-side substituents ligands. This notion also suggests a common framework for designing $S1P_5$-selective ligands.

### Structural insights into inverse agonism

It has been shown that $S1P_5$ exhibits a relatively high level of basal activity[42], while our functional assay revealed that ONO-5430608 acts as an inverse agonist for the $G_i$-protein-mediated signaling pathway, reliably decreasing the basal activity level detected by the BRET-based cAMP sensor (Supplementary Fig. 6).

Our $S1P_5$ structure in complex with the inverse agonist ONO-5430608 along with previously reported agonist and antagonist-bound structures of $S1P_{1,3,5}$ shed light on the mechanism of inverse agonism. Specifically, the above-mentioned conformational flexibility of $Y89^{2.57}$ may provide a structural background for the basal activity of $S1P_5$. We used metaMD to estimate free energy profiles along the reaction coordinate corresponding to the torsion rotation of the $L119^{3.36}$ side chain for $S1P_5$ with $Y89^{2.57}$ restrained in the upward and downward orientations. The upward orientation of $Y89^{2.57}$ is compatible with both active and inactive conformations of the dual toggle switch $L^{3.36}$-$W^{6.48}$, while the downward orientation of $Y89^{2.57}$ selects the inactive conformation (Fig. 3d). The dual toggle switch is found in the previously reported active state structures of $S1P_5$-siponimod as well as in $S1P_1$[23] and $S1P_3$[21] agonist-bound complexes. It induces activation of the P-I-F motif and an outward movement of the intracellular part of TM6 resulting in G-protein signaling cascade. On the other hand, the dual toggle switch is observed in the inactive conformation in our $S1P_5$-ONO-5430608 structure and in the previously published antagonist-bound $S1P_1$[20]. The inverse agonist ONO-5430608 induces the downward conformation of $Y89^{2.57}$ that opens the allosteric subpocket and suppresses the switching of $L119^{3.36}$ locking the dual toggle switch in the inactive state (Fig. 3b). Therefore, the conformational flexibility of $Y89^{2.57}$ in $S1P_5$ provides a structural basis for both receptor subtype selectivity and inverse agonism.

### Naturally occurring mutations in $S1P_5$

In order to characterize additional functionally important residues in $S1P_5$, we performed mapping of known point mutations from genomic databases onto the crystal structure (Fig. 2d). Multiple databases carry information about $S1P_5$ point mutations including gnomAD (229 SNVs)[43], which contains genomic information from unrelated individuals, and COSMIC (124 point mutations)[44], which accumulates somatic mutations in cancer. The most frequent gnomAD mutation

L318[8.55]Q in helix 8 (3% of the population) was shown to impair $G_{12}$ signaling[45]; however, according to our functional data it only slightly decreases the potency of S1P in $G_i$-mediated signaling (Fig. 2c). It was previously proposed[45] that a possible cause of this mutation on the signaling impairment is the prevention of palmitoylation of the downstream C322[8.59] or C323[C-term]. A concomitant cause might be a shift in the helix 8 position due to the loss of a hydrophobic contact between the mutated residue L318[8.55] and the membrane.

Several individuals have missense mutations in the ligand binding pocket; for example, 3 out of 235,080 samples[43] contain R111[3.28]L mutation possibly affecting the contacts with the zwitterionic ligand headgroup (Fig. 2). Mutation of another headgroup recognition residue, E112[3.29]G, is less frequent (1 of 234,568). As shown in our functional data, mutations of both of these residues to neutral ones disrupt response to ligands (Fig. 2c). Additionally, two mutations are located side-by-side in the putative ligand entrance gateway, C43[1.39]F, and M296[7.43]V, are present in the population[43] at $10^{-6}$ frequencies. While C43[1.39]F shows little effect in our functional tests, M296[7.43]V disrupts $G_i$ signaling response for both S1P and ONO-5430608 (Fig. 2c). Another conserved in S1P receptors, except for S1P$_2$, residue A295[7.42] has a hydrophobic contact with the ligand (Fig. 2a), which becomes altered in case of the A295[7.42]S mutation. Mutation of A295[7.42]S may also directly influence the state of the toggle switch (L119[3.36]-W264[6.48] in S1P$_5$) and may interfere with protein activation, as it was shown for several other receptors, e.g., $\beta_2$-adrenergic receptor[46] and CCR5[47]. One of the key residues in the sodium-binding site, N298[7.45], has several variations in population: S, D, or K. While the effects of N298[7.45]D and N298[7.45]S are unclear, N298[7.45]K would mimic sodium-binding, stabilizing the inactive state of the receptor[48].

Somatic mutations appearing in COSMIC and not found in the population may be linked to severe cancer impairments. For example, S125[3.42]R disrupts the conserved hydrogen-bond network involving S77[2.45] and W159[4.50], destabilizing contacts between TMs 2, 3, and 4[49] and, likely, disturbing the 7TM fold due to the introduction of a charged residue in a mostly hydrophobic environment.

### Comparison with AlphaFold predictions

Recently, a redesigned artificial intelligence-based protein structure-predicting system AlphaFold v.2[50] achieved a notable breakthrough in approaching the accuracy in protein structure modeling, previously available only from experimental methods. AlphaFold-based approaches started to find multiple applications in structural biology[51], however, their full capacity and limitations remain to be uncovered. Here, we evaluated the ability of AlphaFold to predict structural features responsible for receptor selectivity and inverse agonism in the S1PR family. For that, we generated 50 de novo AlphaFold models for each of the five S1PRs without using existing structures as templates. Overall, the models demonstrated reasonable correspondence to the available experimental structures; for example, Cα RMSDs in the 7TM region between the S1P$_5$ models and the inactive state crystal structure (S1P$_5$-ONO-5430608) is $1.3 \pm 0.2$ Å and the active state structure (PDB ID 7EW1, S1P$_5$-siponimod) is $3.0 \pm 0.2$ Å.

The conformational heterogeneity of Y(F)[2.57] observed in experimental S1PR structures and metaMD simulations were also well captured by AlphaFold predictions (Fig. 5a–f). In all S1P$_1$, S1P$_3$, and S1P$_4$ models, Y[2.57] has an upward conformation, except for a single S1P$_1$ model, in which this residue adopts a downward orientation similar to that previously observed in all-atom MD simulations[52]. Furthermore, 19 out of 50 S1P$_5$ models display a downward Y[2.57] orientation, while all the others have an upward Y[2.57] orientation. Notably, S1P$_2$ is the only receptor, in which Y[2.57] is replaced with F[2.57] which adopts a downward conformation in all generated models. The downward orientation of F[2.57] in S1P$_2$, similar to that of Y[2.57] in S1P$_5$, opens the allosteric subpocket, which may be targeted to achieve ligand selectivity.

In all available experimental S1PR structures, the conserved dual toggle switch L[3.36]-W[6.48] displays either active or inactive conformation. AlphaFold predicted both of these conformations for all receptors except for S1P$_4$, in which only the active conformation was present in all models (Fig. 5a–e). However, AlphaFold models did not fully reflect the mutual relationship between conformations of Y89[2.57] and L119[3.36], as observed by metaMD in S1P$_5$. Thus, all AlphaFold-predicted S1P$_5$ models cluster into three groups (Fig. 5e), including the energetically unfavorable conformation with Y89[2.57]-L119[3.36] in downward-upward orientations while missing the energetically favorable conformation with Y89[2.57]-L119[3.36] in upward-downward positions. Consequently, we conclude that the current version of AlphaFold could not consistently generate an S1PR structure in a specific signaling state, sometimes mixing the features of different conformations in a single model. These findings are corroborated by a recent study of several other GPCRs[53].

One of the most intriguing AlphaFold-related questions is how useful the predicted models are for structure-based drug design[54]. To test it in application to S1PR targets, we constructed three benchmarks, mimicking virtual ligand screening campaigns, and compare the available experimental structures and AlphaFold models by their ability to distinguish high-affinity ligands from low-affinity binders and decoys. Our results demonstrated that crystal structures outperform AlphaFold-generated models in several scenarios (Fig. 5g, h and Supplementary Fig. 7). Namely, our S1P$_5$ crystal structure showed substantially better overall ranking and top-10% enrichment among both ONO-5430608-like inverse agonists ("ONO" benchmark) and S1P$_5$-selective ligands ("Selective" benchmark). In the case of the non-selective ligand benchmark (mostly S1P$_1$ agonists), the best performance was achieved for several experimental S1PR structures determined in complex with non-selective ligands, e.g., S1P$_1$-siponimod complex (Fig. 5g, h), while our S1P$_5$ structure fared on par with AlphaFold models.

## Discussion

Here, we present the 2.2 Å crystal structure of the human S1P$_5$ receptor in complex with its selective inverse agonist. The structure was obtained by room temperature SFX data collection at PAL-XFEL using sub-10 μm crystals. In combination with site-directed mutagenesis, functional assays, metaMD simulations, and docking studies, this structure revealed molecular determinants of ligand binding and selectivity as well as shed light on the mechanism of inverse agonism in the S1PR family. The obtained structure also allowed us to map locations of known missense SNVs from gnomAD and COSMIC genome databases and annotate their potential functional roles providing future insights into personalized medicine approaches.

We found that the inverse agonist ONO-5430608 binds to the receptor's orthosteric site, suppressing S1P$_5$ basal activity. Highly conserved residues Y19[N-term], K24[N-term], R111[3.28], and E112[3.29] play an essential role in the recognition of both ONO-5430608 and its native ligand S1P. The naphthyl group of ONO-5430608 occupies an allosteric subpocket that was not previously observed in any other S1PR structure. While the orthosteric site is highly conserved in the S1PR family, the allosteric subpocket is composed of unique residues and is present in our S1P$_5$ structure due to the conformational switch of a single residue Y[2.57]. Functionally important residues were revealed by structure-guided site-directed mutagenesis and $G_i$ signaling assays. We further used metaMD simulations to explore the conformational flexibility of Y[2.57] in S1PRs and established its role in receptor subtype selectivity and inverse agonism. The role of Y[2.57] in the binding of selective ligands was also confirmed by comparative molecular docking simulations. Furthermore, taking advantage of the availability of several experimental structures of S1PRs in different functional states, we tested the ability of AlphaFold to predict de novo specific conformational states for S1PRs and to provide reliable templates for structure-based virtual ligand screening. While the AlphaFold-

generated models showed a close similarity to experimental structures and captured conformational diversity of conserved structural motifs, the models did not provide a full description of specific signaling states and showed subpar performance in virtual ligand screening compared to experimental structures.

Our structure along with our functional and computer modeling data may facilitate the rational design of ligands that could further serve as lead or tool compounds for detailed elucidation of biological function of S1P$_5$ and therapeutic developments. S1P$_5$ is emerging as a promising drug target. Inhibiting S1P$_5$ by an inverse agonist could create new therapeutic strategies against neuroinflammation and degeneration where the high ligand selectivity would diminish the off-target effects. While S1P$_1$ has a broad expression profile, S1P$_5$ is expressed predominantly in brain tissues[8]; thus, a highly selective compound would afford more localized control over associated CNS disorders not affecting peripheral processes in the body.

## Methods

### Protein engineering for structural studies

The human wild-type gene *S1PR5* (UniProt ID Q9H228) was codon-optimized by GenScript for insect cell expression and modified by adding a hemagglutinin signal peptide (HA; KTIIALSYIFCLVFA), a FLAG-tag for expression detection, and an Ala-Gly-Arg-Ala linker at the N-terminus. An apocytochrome b562RIL (BRIL)[26] was inserted in the third intracellular loop between A223 and R241 to stabilize the receptor and facilitate crystallization. The C-terminus was truncated after Val321, and a PreScission cleavage site was added after it to enable the removal of the following 10× His tag used for IMAC purification (Supplementary Fig. 1). The resulting construct was cloned into a pFastBac1 (Invitrogen) plasmid. The full DNA sequence of the S1P$_5$ crystallization construct is provided in Supplementary Table 3.

### Protein expression

Using the Bac-to-Bac system (Invitrogen), a high titer ($10^9$ particles per ml) virus encoding the crystallization construct was obtained. Sf9 (Novagen, cat. 71104) cells were infected at a density $(2-3) \times 10^6$ cells per ml and a multiplicity of infection (MOI) 4-8, incubated at 28 °C, 120 rpm for 50-52 h, harvested by centrifugation at 2,000×*g* and stored at −80 °C until further use.

### Protein purification

Cells were thawed and lysed by repetitive washes (Dounce homogenization on ice, and centrifugation at 128,600×*g* for 30 min at 4 °C) in hypotonic buffer (10 mM HEPES pH 7.5, 20 mM KCl, and 10 mM MgCl$_2$) and high osmotic buffer (10 mM HEPES pH 7.5, 20 mM KCl, 10 mM MgCl$_2$, and 1 M NaCl) with an addition of protease inhibitor cocktail [PIC; 500 µM 4-(2-aminoethyl)benzenesulfonyl fluoride hydrochloride (Gold Biotechnology), 1 µM E-64 (Cayman Chemical), 1 µM leupeptin (Cayman Chemical), 150 nM aprotinin (A.G. Scientific)] with the ratio of 50 µl per 100 ml of lysis buffer. Membranes were then resuspended in 10 mM HEPES pH 7.5, 20 mM KCl, 10 mM MgCl$_2$, 2 mg/ml iodoacetamide, PIC (100 µl per 50 ml of resuspension buffer), and 50 µM ONO-5430608 (4-{6-[2-(1-Naphthyl)ethoxy]−1,2,4,5-tetrahydro-3H-3-benzazepin-3-yl}butanoic acid; Example 18(18)[25], received as a gift from Ono Pharmaceutical) for 30 min at 4 °C and then solubilized by addition of 2× buffer (50 mM HEPES, 500 mM NaCl, 2% w/v n-dodecyl-β-D-maltopyranoside (DDM; Anatrace), 0.4%w/v cholesteryl hemisuccinate (CHS; Sigma), 10%v/v glycerol) and incubation for 3 h at 4 °C with 10 rpm rotation. All further purification steps were performed at 4 °C. The supernatant was clarified by centrifugation (292,055×*g*, 60 min, 4 °C) and bound to 2 ml of TALON IMAC resin (Clontech) overnight with 10 rpm rotation in the presence of 20 mM imidazole and NaCl added up to 800 mM. The resin was then washed with ten column volumes (CV) of wash buffer I (8 mM ATP, 50 mM HEPES pH 7.5, 10 mM MgCl$_2$, 250 mM NaCl, 15 mM imidazole, 50 µM ONO-

5430608, 10%v/v glycerol, 0.1/0.02%w/v DDM/CHS), then with five CV of wash buffer II (50 mM HEPES pH 7.5, 250 mM NaCl, 50 mM imidazole, 50 µM ONO-5430608, 10%v/v glycerol, 0.5/0.01%w/v DDM/CHS), then eluted with (25 mM HEPES pH 7.5, 250 mM NaCl, 400 mM imidazole, 50 µM ONO-5430608, 10%v/v glycerol, 0.05/0.01%w/v DDM/CHS) in several fractions. Fractions containing target protein were desalted from imidazole using PD10 desalting column (GE Healthcare) and incubated with 50 µM ONO-5430608 and a His-tagged PreScission protease (homemade) overnight with 10 rpm rotation to remove the C-terminal 10× His tag. Protein was concentrated up to 40−60 mg/ml using a 100 kDa molecular weight cutoff concentrator (Millipore). The protein purity was checked by SDS-PAGE. Yield and monodispersity were estimated by analytical size exclusion chromatography. Stability and stabilizing effect of the ligand were measured by microscale fluorescent thermal stability assay[55] (Supplementary Fig. 2).

### Thermal stability assay

Microscale fluorescent thermal stability assay[55] was conducted using a CPM dye (7-Diethylamino-3-(4-maleimidophenyl)-4-methylcoumarin, Invitrogen) dissolved in DMF at 10 mM. This CPM stock solution was diluted to 1 mM in DMSO and then added to working buffer at 10 µM. 1 µg of the target protein was added to 50 µL of working buffer (25 mM HEPES, 250 mM NaCl, 10%v/v glycerol, 0.05%w/v DDM, 0.01%w/v CHS) with CPM, and the melting curve was recorded on a Rotor-Gene Q real-time PCR cycler (Qiagen) using a temperature ramp from 28 to 98 °C with 2 °C/min rate. The fluorescence signal was measured in the Blue channel (excitation 365 nm, emission 460 nm), and the melting temperature was calculated as the maximum of the fluorescence signal derivative with respect to temperature.

### LCP crystallization

Purified and concentrated S1P$_5$ was reconstituted in LCP, made of monoolein (Nu-Chek Prep) supplemented with 10%w/w cholesterol (Affymetrix), in 2:3 (v/v) protein:lipid ratio using a syringe lipid mixer[27]. The obtained transparent LCP mixture was dispensed onto 96-wells glass sandwich plates (Marienfeld) in 40 nl drops and covered with 900 nl precipitant using an NT8-LCP robot (Formulatrix) to grow crystals for synchrotron data collection. To prepare crystals for XFEL data collection, the protein-laden LCP mixture was injected into 100 µl Hamilton gas-tight syringes filled with precipitant as previously described[27]. All LCP manipulations were performed at room temperature (20−23 °C), while plates and syringes were incubated at 22 °C. Crystals of S1P$_5$ grew to their full size of <30 µm (in plates) or <10 µm (in syringes) within 3 days in precipitant conditions containing 100−300 mM KH$_2$PO$_4$ monobasic, 28−32%v/v PEG400, and 100 mM HEPES pH 7.0.

### Diffraction data collection and structure determination

XFEL data for S1P$_5$-ONO-5430608 crystals were collected at the NCI (Nanocrystallography and Coherent Imaging) beamline of the Pohang Accelerator Laboratory X-ray Free Electron Laser (PAL-XFEL), Pohang, South Korea. The PAL-XFEL was operated in SASE mode at the wavelength of 1.278 Å (9.7 keV) and 0.2% bandwidth, delivering individual X-ray pulses of 25-fs duration focused into a spot size of 2 × 3 µm using a pair of Kirkpatrick-Baez mirrors. LCP laden with dense suspension of protein microcrystals was injected at room temperature inside a sample chamber filled with helium (23 °C, 1 atm) into the beam focus region using an LCP injector[56] with a 50-µm-diameter capillary at a flow rate of 0.15 µl/min. Microcrystals ranged in size from 5 to 10 µm. Diffraction data were collected at a pulse repetition range of 30 Hz with a Rayonix MX225-HS detector, operating in a 4 × 4 binning mode (1440 × 1440 pixels, 30 fps readout rate). The beam was not attenuated and delivered full intensity ($5 \times 10^{11}$ photons per pulse). A total number of 490,000 detector images were collected. Due to a high systematic background, Cheetah[57] v. 2019-1 was initially used only to apply dark

current calibration, and all images were used for further processing. The overall time of data collection from a sample with a total volume of about 36 μl was approximately 4 h and yielded 6918 indexed frames with 7492 crystal lattices.

During the XFEL data collection, a high systematic background scattering from upstream to the interaction point occurred due to a high-intensity X-ray lasing conditions (Supplementary Fig. 4; Matplotlib[58] v.3.3.2 was used for radial averaging of the scattered intensity), which prevented from establishing suitable Cheetah hit finding parameters during the beamtime and complicated the overall data processing. All data processing was performed using CrystFEL[59] v. 0.8.0. Here we describe steps that we took to improve data quality as much as possible starting from the available data with a high background level. For all CrystFEL runs (Supplementary Table 4), peak search was limited with max-res = 340, min-res = 50 to search for peaks in the region between the beamstop and the LCP ring, and the frames were limited to a 12,000 subset of all frames, selected with minimum 5 peaks with SNR 2.7. Initially, typical starting peak finding parameters (SNR = 5.0, threshold = 100) in CrystFEL were used for data processing, yielding only 2036 crystals with indexing = mosflm,dirax,xgandalf (Supplementary Table 4 column A). Initial peak search parameter adjustment, as described in CrystFEL tutorial[59], led to the value of SNR = 2.7 and threshold = 30, which yielded 5275 crystals (Supplementary Table 4 column B). Applying -median-filter = 5 allowed to further increase the number of crystals to 7189, while increasing SNR to 4.0 (Supplementary Table 4 column C).

Spot integration parameters had the biggest impact on the merged data quality. First, changing the spot integration model from rings-nograd model, which assumes flat background around a spot, to rings-grad, which performs 2D-fitting of each spot background profile, decreased overall $R_{split}$ from 29.7% to 19.4% (Supplementary Table 4 column D) and increased the highest resolution shell CC* from 0.618 to 0.666. Second, increasing local-bg-radius from 3 to 5, and using int-radius = 3,5,8 instead of default 4,5,8 further improved data quality with the highest resolution shell CC* equal to 0.716 (Supplementary Table 4, columns E-F). Following reviewer's suggestions, we attempted to improve overall data resolution via applying partiality modeling (column C'), less aggressive push-res option with or without–overpredict option (columns H and I, respectively). None of these strategies yielded better results than the initial processing (column G). The final merging was performed with partialator, iterations = 2, push-res = 5.0, and model = ggpm (Supplementary Table 1).

The structure was initially solved by molecular replacement using phenix.phaser[60] with two independent search models of the poly-alanine S1P$_1$ 7TM domain (PDB ID 3V2Y) and BRIL from the high-resolution A$_{2A}$AR structure (PDB ID 4EIY). Model building was performed by cycling between manual inspection and building with Coot[61] v. 0.9.6 using both 2$m$Fo-$D$Fc and $m$Fo-$D$Fc maps and automatic refinement with phenix.refine[62] v. 1.19.2 using automatic torsion angle NCS restraints and 2 TLS groups. Ligand restraints were generated using the web server GRADE v. 1.2.19 (http://grade.globalphasing.org). The S1P$_5$ structures from two molecules A and B in the asymmetric unit show very high similarity (Ca RMSD 1.0 Å within 7TM; 1.3 Å all-atom RMSD). The main difference includes flexible ECL1 and conformations of several side chains exposed to the lipid bilayer and solvent. The final data collection and refinement statistics are shown in Supplementary Table 1. The relatively high Rfree of the structure can be partially explained by the high systematic background scattering and modulations of the diffraction intensities. The modulations are produced by two factors: (1) the NCS operator (x, y, z) → (1/8 + x, -y, -z) seen as a Patterson peak at (3/8, 1/2, 0) with a 0.3 of the origin peak height, and (2) the lattice-translocation defect (LTD)[63] seen as a Patterson peak at (1/4, 0, 0) with a 0.1 of the origin peak height. We corrected our data partially for LTD as described previously[64] which resulted in a Rfree

drop by 0.6% during the refinement. The final resolution cutoff was determined by paired refinement[65].

## AlphaFold predictions
Prediction runs were executed using AlphaFold[50] v. 2.1.1 + 110948 with a non-docker setup (https://github.com/kalininalab/alphafold_non_docker, git commit 7ccdb7) and an updated run_alphafold.sh wrapper with added -random-seed parameter. The use of structural templates was disabled by setting "max_template_date" to 1900-01-01; thus, no S1PR structures were used for prediction, and all AlphaFold models analyzed in this work were constructed based on multiple sequence alignment alone. 50 AF2-models (ranked_....pdb models) were generated for each 5 human S1PRs with protein sequences obtained from UniProt. For each receptor, 10 prediction runs with different seeds (-random-seed" = <run_number>) were executed; each run generated five models. Structures were used as provided by the Alphafold's pipeline with Amber relaxation (see Supplementary Methods 1.8.6 in Ref. 51 for details) without any further modifications (Supplementary Data file 1).

## MD simulations
Molecular dynamics simulations were conducted for the wild-type human S1P$_1$, S1P$_3$, and S1P$_5$ receptors based on the X-ray structures 3V2Y[20] (residues V16-K300), 7C4S[21] (G14-R311), and the structure reported in the present study (S12-C323), respectively. All engineered mutations were reverted back to the WT amino acids, and all missing fragments were filled using Modeller[66] v. 9.24. Receptors were embedded into lipid bilayers consisting of 1-palmitoyl-2-oleoyl-sn-glycero-3-phosphatidylcholine (POPC) lipids and solvated with TIP3P waters and Na$^+$/Cl$^-$ ions (to guarantee the electroneutrality of the systems and the ionic strength of 0.15 M) by means of the CHARMM-GUI web-service[67]. The obtained in this way starting models (with 61,666/61,763/56,303 atoms including 119/117/123 POPC molecules in the S1P$_1$/S1P$_3$/S1P$_5$ systems, respectively) were subject to standard CHARMM-GUI minimization and equilibration protocol, i.e., the steepest descent minimization (5000 steps) was followed by a series of short equilibration simulations in the NPT ensemble using Berendsen thermostat and barostat with the restraints on protein and lipids gradually released.

We employed a metadynamics (metaMD) approach[41] to estimate free energy profiles along the rotation of the $\chi_1$ torsion angle in the side chain of Y$^{2.57}$ in S1P$_1$, S1P$_3$, and S1P$_5$ as well as free energy profiles along the rotation of the $\chi_1$ torsion angle in the side chain of L$^{3.36}$ in S1P$_5$ with two alternative orientations of Y$^{2.57}$. This method is based on the addition of biasing repulsive potentials ("hills", typically Gaussians) to the total potential of the system to enhance the sampling of the configurational space along the chosen reaction coordinates. The deposition rate for hills in metaMD simulations was 1 ps$^{-1}$; the width and height of deposited hills were equal to 0.1 rad (-5.7°) and 0.5 kJ/mol, respectively. The metaMD simulations were run for 10 ns each. Two conformations corresponding to the free energy minima along the rotation of the $\chi_1$ torsion of Y$^{2.57}$ in S1P$_5$ were selected for the subsequent metaMD simulations of L$^{3.36}$, in which the orientation of Y$^{2.57}$ was harmonically restrained in the upward or downward positions. To test for convergence of the metaMD simulations, we applied the following method[68]: the free energy difference between two regions of the obtained free energy profiles, corresponding to the crystal-lographic orientations of Y$^{2.57}$ (Supplementary Fig. 8a–c) or to the orientations of L$^{3.36}$ in the active and inactive S1P$_5$ structures (Supplementary Fig. 8d, e) as a function of simulation time was plotted. In case of convergence, this difference should not change with the progress of simulations as the systems diffuse freely along the reaction coordinate.

For the metaMD simulations, Nose–Hoover thermostat and Parrinello–Rahman barostat were used. The temperature and pressure were set to 323.15 K and 1 bar with temperature and pressure coupling

time constants of 1.0 ps⁻¹ and 0.5 ps⁻¹, respectively. All MD simulations were performed with GROMACS[69] v. 2020.2 using PLUMED plugin[70] to enable metaMD. The time step of 2 fs was used for all production simulations. The CHARMM36 force field[71] was used for the proteins, lipids, and ions.

## SAR and molecular docking

For $S1P_5$ docking studies, we used chain B from our $S1P_5$-ONO-5430608 crystal structure and a metaMD snapshot with the upward conformation of $Y89^{2.57}$. Chain B was selected based on the quality of $2mFo$-$DFc$ maps around the ligand and surrounding residues. Molecular docking was performed using ICM Pro v. 3.9-1b (Molsoft, San Diego). We removed ligands and converted the receptor models into an ICM format using default settings, which includes building missing side chains, adding hydrogens, energy-based Gln/Asn/His conformation optimization, and removal of all water molecules. The same docking box was selected for both models, aligned by their 7TM domains, to encompass both orthosteric and allosteric binding pockets. For each ligand we repeated docking runs 5 times with the effort parameter (ligand sampling depth) set at 16, each time saving three best conformations. Ligand structures and their affinities ($IC_{50}$ values from radioligand binding assays) at $S1P_5$ receptors were taken from the published patent[25].

In the AlphaFold models analysis, 50 $S1P_5$ models predicted by the AlphaFold algorithm were compared with both chains of our $S1P_5$ crystal structure and other available S1PR crystal structures. All structures were prepared as described above. $S1P_5$ ligands from ChEMBL[72] v. 29 were accessed via the web-interface (https://www.ebi.ac.uk/chembl/) using the $S1P_5$'s ChEMBL target ID. Ligands were converted to 3D and charged at pH 7.0 using Molsoft ICM. For each model, ligand screening was performed three times with docking effort 1. Three ligand benchmarks (Supplementary Fig. 7) were used: 1. "ONO" series: active molecules from ref. 25 (group A, 1 nM < $IC_{50}$ < 100 nM), inactive molecules from Ref. 25 (group C, 1 µM < $IC_{50}$ < 3 µM) and decoys; 2. "Selective" series: active molecules from refs. 25,73 (group A or $IC_{50}$ < 100 nM, correspondingly), inactive molecules from refs. 25,73 (group C or $IC_{50}$ >= 1 µM, correspondingly), and decoys; 3. "Non-selective" series: active molecules from ChEMBL (pChembl >7.0, mostly $S1P_1$ agonists), inactive molecules from ChEMBL (pChembl <5.0), and decoys. Decoy molecules were selected from the Enamine REAL library [https://enamine.net/compound-collections/real-compounds/real-database], matching the distribution of active molecules by charge and weight. The benchmarks have the following ratios of active:inactive:decoy molecules: 6:5:60 for "ONO", 12:10:120 for "Selective", and 158:39:1207 for "Non-selective", with the imbalance parameter (ratio of the total library size to the number of active molecules in it) of 11.8, 11.8, and 8.8, respectively. Docking scores and ligand structures are provided in Supplementary Data file 2. For estimation of the virtual screening quality, metrics enrichment at 10% and receiver operating characteristic (ROC)−area under the ROC curve (AUC) were used, as implemented in RDKIT[74] v. 2021-03-4. Data were plotted using Seaborn[75] v.0.11.1 with Matplotlib[58] v.3.3.2.

## Plasmids for functional assays

The human wild-type *S1PR5* gene (UniProt ID Q9H228) with an N-terminal 3× HA epitope (YPYDVPDYA) tag was cloned into pcDNA3.1(+) (Invitrogen) at KpnI(5′) and XhoI(3′). Point mutations were introduced by overlapping PCR. All DNA sequences were verified by Sanger sequencing (Evrogen JSC). Sequences of all primers used in this work are listed in Supplementary Table 5.

## Cell surface expression determined by ELISA

Cell surface expression of $S1P_5$ receptor variants was determined by whole-cell ELISA[76]. Briefly, HEK293T cells were seeded in 24-well cell culture plates (0.2×10⁶ cells in 0.5 ml of medium per well) and

transfected separately by 3 µg of each expression plasmids based on pcDNA3.1(+) vector using common Lipofectamine 3000 protocol. After 12−18 h incubation in a $CO_2$ incubator at 37 °C for receptor expression, the cell culture plates were placed on ice, the media was aspirated completely, and the cells were washed once with ice-cold TBS to remove any residual media. Then the cells were fixed using 400 µl of 4%w/v paraformaldehyde, followed by three 400−500 µl washes with TBS. After surface blocking with 2%w/v protease-free BSA (A3059, Sigma) solution in TBS, HRP-conjugated anti-HA high affinity antibody (3F10) (Roche) at a dilution of 1:2000 in TBS + 1%w/v protease-free BSA and TMB ready-to-use substrate (T0565, Sigma) were used for ELISA procedure. The ELISA results were normalized by Janus Green staining. Cells transfected with empty vectors (pcDNA3.1(+)) were used to determine background.

## Functional assays with BRET-based cAMP sensor

$G_i$ protein-mediated signaling responses to endogenous agonist S1P and inverse agonist ONO-5430608 were assayed for human WT and mutant $S1P_5$ receptors using Bioluminescence Resonance Energy Transfer (BRET) based EPAC biosensor[77]. Briefly, transfections were carried out by Lipofectamine 3000 according to standard protocol using HEK293T cells seeded in a 100 mm cell culture plate, receptor cDNA vectors (10 µg each), and EPAC biosensor cDNA vector (10 µg) needed for evaluation of cAMP production. Transfected cells were split into 96-well plates at 10⁵ cells per well and incubated for 16−18 h. To measure response for S1P, 60 µl of PBS was added to each well followed by addition of 10 µl of a 50 µM coelenterazine-h, 10 µl of 300 µM forskolin and 10 µl of 100 µM 3-isobutyl-1-methylxanthine (IBMX) solutions. After 10-min incubation, either 10 µl of vehicle or 10 µl of S1P at different concentrations in 0.5%w/v fatty acid-free BSA (10775835001, Roche) solution in PBS was added. To measure response for ONO-5430608, 70 µl of PBS was added to each well followed by addition of 10 µl of 50 µM coelenterazine-h and 10 µl of 100 µM IBMX solutions. After 10-min incubation, either 10 µl of vehicle or 10 µl of ONO-5430608 at different concentrations in PBS was added. The plate was then placed into a CLARIOstar reader (BMG LABTECH, Germany) with a BRET filter pair (475 ± 30 nm−coelenterazine-h and 550 ± 40 nm−YFP). The BRET signal was determined by calculating the ratio of the light emitted at 550 nm to the light emitted at 480 nm. The $EC_{50}$ values were calculated using the three-parameter dose−response curve fit in GraphPad Prism v. 9.3. Three independent experiments were performed in triplicate.

## Reporting summary

Further information on research design is available in the Nature Research Reporting Summary linked to this article.

# Data availability

Coordinates and structure factors for the $S1P_5$-ONO-5430608 structure have been deposited in the Protein Data Bank (PDB) under the accession code 7YXA. Raw SFX diffraction data have been deposited to CXIDB database under accession number 196. Publicly available amino acid sequences for S1PRs used in this study were obtained from the UniProt database under accession numbers: P21453, O95136, Q99500, O95977, Q9H228. Publicly available structures used in this study can be found in the Protein Data Bank under accession codes: 3V2W, 3V2Y, 4EIY, 7C4S, 7EVY, 7EW1, 7EW2, 7EW4. SNV data for $S1P_5$ used in this work are available from public databases gnomAD [https://gnomad.broadinstitute.org/gene/ENSG00000180739?dataset=gnomad_r2_1] and COSMIC [https://cancer.sanger.ac.uk/cosmic/gene/analysis?ln=S1PR5]. AlphaFold structures, sequences, and scripts used to generate them are provided in Supplementary Data file 1. Structures of the compounds used for docking to experimental and AlphaFold

structures and their docking scores are provided in Supplementary Data file 2. Source Data are provided in this paper.

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

## Acknowledgements

We thank S. Ustinova, A. Podzorov, A. Awawdeh, P. Utrobin, and Yu. Semenov for technical assistance, T. Maruyama for providing ONO-5430608. We acknowledge the Paul Scherrer Institut, Villigen, Switzerland for the provision of synchrotron radiation beamtime at beamline PX1 of the SLS and would like to thank Dr. V. Olieric for assistance. The authors are grateful to the staff of the accelerator and beamline departments at PAL-XFEL for their technical support. The XFEL experiments were performed at the NCI PAL-XFEL experimental station under proposal No. 2019-2nd-NCI-012. Protein production and crystallization were supported by the Ministry of Science and Higher Education of the Russian Federation agreement 075-00337-20-03, project FSMG-2020-0003 (V.B., A.M., E.L., A.Gu., M.S., A.L.). XFEL and synchrotron data collection and treatment were supported by the Russian Ministry of Science and Higher Education grant No. 075-15-2021-1354 (V.B., A.M., E.M., E.L., A.R.). SFX data collection strategy was developed with the support from the Russian Foundation for Basic Research (RFBR) project 18-02-40020 (V.B., A.M., E.M.). Functional assays were developed and implemented with the support of the Russian Science Foundation project 22-74-10036 (A.M., A.L., E.L.). Molecular docking simulations were supported by the Russian Science Foundation project 22-24-00454 (E.M., M.K.). J.P. and Y.C. were supported by the National Research Foundation of Korea (grant No. NRF-2017M3A9F6029736). U.W. is supported by the National Science Foundation (NSF) BioXFEL Science and Technology Center award 1231306. W.L. is supported by the Advancing Healthier Wisconsin Endowment (AHW) fund. The authors are grateful to the Global Science Experimental Data Hub Center (GSDC) for data computing and the Korea Research Environment Open NETwork (KRE-ONET) for network service provided by the Korea Institute of Science and Technology Information (KISTI) and the Data Processing Center of Moscow Institute of Physics and Technology for high-performance data computing infrastructure and technical support.

## Author contributions

E.L. and A.Gu. optimized the constructs, developed the expression and purification procedure, expressed and purified the protein, screened the ligands, and crystallized the protein–ligand complexes. E.M., D.V., A.M., and V.C. collected X-ray diffraction data at PAL-XFEL. J.P. set up the XFEL experiment, beamline, controls, and data acquisition; operated the beamline. H.H., U.W. helped develop and operate the LCP injector. W.L. helped with the XFEL sample preparation. S.P. contributed to the

experimental system installation at the beamline. G.P. contributed to DAQ and data handling. H.J.H. contributed to the Rayonix detector installation and operation. A.Gu., E.M., E.L., K.K, S.B. and A.M. collected synchrotron data at SLS. A.Ge. performed and analyzed cell signaling and cell surface expression assays. E.M., D.V., and V.B. processed diffraction data. G.B. helped with X-ray diffraction data interpretation and analysis. E.M., V.B., V.C. performed structure determination and refinement. V.B., V.C., E.L., A.Gu., E.M., A.M., A.L. performed project data analysis/interpretation. M.E., A.L., G.K., and M.S. helped with construct optimization, protein expression, and purification. P.P. advised on the protein construct design. E.M., M.K. performed molecular docking. P.O., E.M. performed MD simulations. E.M., I.G., V.B. performed AlphaFold simulations and their data analysis. A.L., I.O., P.Kh., A.R., Y.C. helped with experimental work and project organization. E.L., E.M., A.Gu., P.O., A.M., V.B., V.C., wrote the manuscript with the help from other authors. V.C, A.M., V.B., and V.G. initiated the project. A.M. and V.B. organized the project implementation, were responsible for the overall project management and co-supervised the research. V.C. supervised the overall project.

## Competing interests

The authors declare no competing interests.
