## [Peer Review File · Nature Communications]

Structural basis for receptor selectivity and inverse agonism in S1P5 receptorsREVIEWER COMMENTS

Reviewer #1 (Remarks to the Author):

In the manuscript, Lyapina and et al., obtained a crystal structure of the S1P5 receptor in complex with ONO-5430608 and analyzed this structure involving mutagenesis and molecular modeling. The structure shows that ONO-5430608 binds partially to the allosteric cavity composed of non-conserved residues. This cavity is opened up due to the rotation of the Y89 side chain. This binding site has not been seen in other available structures of S1PRs and this is a major highlight of the story, and it could advance the field. Next, the authors compare the structure with other available structures of the S1PR family and link it with functional motifs. They also conducted a range of modelling exercises to see the rotamer states of Y89, and binding of analogs and compared them with AlphaFold structures by performing virtual screening. In addition, the structure was used to map various naturally occurring, missense and disease-related mutations from various databases. The computational part of the work provides further validation of the experimental structure. Overall, this is an interesting study and has a good potential to be published in the journal if the authors address the following comments:

1. I had difficulties following the findings as most of them are hidden in the supporting info. I recognize there might be limits on the number of figures in the main text but the authors should try to put a figure for each part of the study. Currently, the text is disproportional to the available figures. Large sections of the text (docking, AlphaFold, SNPs) are not linked to any figures. It might be good to reduce or remove these sections (see below).

1. 'however, the first two residues of the P-I-F motif deviate from the consensus' – this part of the sentence should be rephrased, it gives confusion.

1. Line 219-222. Please check these two sentences. The text contradicts Fig 2c. L292A/V has a small change in the activity compared to the wild type, whereas M296/V/W has no response. G293A has minor changes in the activity, while G293V has no response.

1. Lines 248-272: The section of docking should be shortened or moved to the supporting info as it doesn't give much new info about the structure but further tests the position of Y89. Description of groups for ligands based on the activity without showing the chemical structures is challenging to follow. The reference to the paper doesn't help. The authors may want to give the structures of all the groups in the supporting info. The authors give docking scores without any units, which also gives the challenge to appreciate the importance of the scoring values. Again, the SAR analysis discussion at the end of the section is difficult to grasp without the structures. Given that the authors don't validate the docking by

completing mutagenesis for the selected analogs and rely mainly on the docking scores I would really recommend moving this section to the supporting info.

1. Lines 273-291: This section is very speculative and should be removed from the manuscript. It is not clear from the structure and metadynamics performed that Y89 is responsible for the inverse agonism and linked to other activation motifs such as L3.36-W6.48.

1. Line 292-321: Supporting Figure 9 should be in the main text.

1. Line 322-363: the authors should comment on whether any structures of the S1PR family were in the training set of AlphaFold. The results of VS should be in the main text.

Reviewer #2 (Remarks to the Author):

Please see attached file, 'Comments_4_2_22.docx'.

In the manuscript ‘Structural basis for receptor selectivity and inverse agonism in S1P₅ receptors’, Lyapina et al describe the crystal structure of an inverse agonist-bound S1P₅ receptor, a GPCR in the class A family. Although many class A GPCR structures have been solved, including some in the S1P receptor subfamily, the new structure presented in this manuscript provides valuable insights into the inverse agonism and ligand selectivity of S1P₅ receptors. The authors validated their structural findings with mutational analyses, molecular dynamics simulation, and molecular docking. Furthermore, they compared the available experimental structures with AlphaFold models of S1PR5 in their ability to distinguish S1PR ligands based on affinity and identify potential drug candidates. This study is mostly complete, and there are only a few concerns that need to be addressed.

1. The authors indicate that the rotamer conformation of Y89^{2.57} is correlated with the activation state of the receptor, and this aromatic residue serves as a selectivity determinant for S1P₅-specific ligands. In the known structure of agonist siponimod-bound S1P₅ receptor, Y89^{2.57} assumes an upward orientation, but it does not make direct contact with the agonist (Yuan et al., Cell Research, 2021). According to the functional assay conducted in this study, the Y89A or Y89W mutations show similar effects on the potencies of S1P agonism and ONO-5430608 antagonism. Could the authors provide an explanation for these observations?

2. In the structure of ONO-54306-bound S1P₅ receptor, Y89^{2.57} forms direct contact with the tetrahydro-benzazepine double ring of ONO-5430608. Could the downward conformation of Y89^{2.57} be induced by the inverse agonist ONO-5430608?

Furthermore, all the ligands in Fig. S8 have a core double ring but different substituents that correspond to different receptor-binding affinities. Could the core double ring system determine inverse agonism of these ligands by forcing a downward orientation of Y89^{2.57}?

3. The authors indicated that the mutational effects of residues lining the allosteric subpocket are correlated with their occluded area while interacting with ONO-543060. The residue I93^{2.61} has a large occluded area (127 Å) when ONO-543060 is bound to the receptor, similar to L292^{7.39} and M296^{7.43}. Why wasn't this residue tested in functional assays?

4. Some naturally occurring mutations such as C43F in the putative ligand entrance gateway, show little effect in functional tests. Could the authors speculate what might be the underlying reason for this?

5. Fig. S2a shows that the purified S1P₅ receptor protein is still quite heterogeneous based on results from analytical size exclusion chromatography. Why wasn't size exclusion chromatography applied on preparative scale to further purify the protein?

6. The authors indicated that the EC50 value of ONO-5430608 in Gi-mediated cAMP accumulation assay is 1.7nM (page 6 under the section ‘overall structure of the ligand-binding pocket’). This is different from the data reported in Table S2 (pEC50=8.54M). Please clarify the difference. Was this value derived experimentally or cited from literature?

Figures:

1. The residues in Fig. 3a are not correctly labeled. The S1P₁ receptor should have M124^{3,32} and V301^{7,43}. The S1P₅ receptor has V115^{3,32} and M296^{7,43}.
2. The authors may consider adding a panel in Fig. 1 that shows only the new structure described in this manuscript.

Reviewer #3 (Remarks to the Author):

Key results

The authors have used serial femtosecond crystallography at the Korean XFEL in order to obtain the structure of a sub-type 5 sphingosine-1-phosphate GPCR in complex with an inverse agonist. While several structures of the same family have been obtained and a structure of the sub-type 5 in complex with an agonist has been recently reported, there was no structure with an inverse agonist available. The authors use several techniques ranging from x-ray crystallography to activity assays of multiple mutants designed based on the crystal structure over MD simulations to generating structures using AlphaFold and comparing them to their obtained structure and other structures. Furthermore, the authors performed molecular docking simulations and mapped naturally occurring mutations onto their structure. They identify a tyrosine residue that appears to play a critical role in sub-type specificity of the inhibitor the structure was obtained with and subsequently analyze its role within the receptor family using the techniques mentioned above.

Validity

Since my expertise is mostly in crystallography and serial crystallography, I can judge the data validity best in these areas. The crystallographic interpretation of the data is solid, despite the high R-factors (see quite a lot of comments on that below that may help to fix this). There are minor questions remaining about the role and modeling of the sodium binding site (see below). The analysis of the auxiliary data appears to be sound and the methods are sufficiently detailed for a non-expert to understand the way they were implemented.

Significance

The structure presented in this work is one of the highest resolution GPCR structures available to date and is one of the few that have been determined at physiological temperatures. It is furthermore the first GPCR structure to be determined at the PAL-XFEL, a new free electron laser which is not long in full operation and will therefore get attention from the field. The newly identified sub-pocket and the carefully analyzed role of Tyr 89 in the context of the other four isoforms shed an interesting light on the structural plasticity of this protein. The analysis of the structure in context of the AlphaFold generated structures shows the promise but also the limitations of modern structure prediction, especially the finding that predicted models can produce distinct, but also mixed signaling states resulting in an underperformance in molecular docking when compared to crystallographic structures.

I am a little bit weary of the “drug design” direction presented, since no clear recommendations or conclusions are drawn from the large amount of data. I feel a functional analysis of the receptor based on the data obtained by the authors would have suited the reader more. If there is drug development potential in this structure it is certain that it will be used and even more so in the light of the amount of auxiliary data the authors presented. But given the authors competence in the GPCR field it feels like there is more functional insight to this data that an in-depth analysis may have revealed.

Point-to-point response:

Main manuscript:

Page 4:

“Attempts to solve the structure were first made using cryocooled crystals at a synchrotron source achieving a maximum of 4 Å single-crystal data resolution, however, the obtained data could not be phased using molecular replacement.”

The reader does not know anything about this experiment and how seriously it was attempted, given a crystal size of 30 µm it is clear that the dose should have been spread over multiple crystals, which is standard for GPCR cryo-crystallography at synchrotrons. The passage creates the impression that a much higher resolution was obtained at the XFEL without doing an experiment that would show this, is therefore misleading and should be removed.

“The crystal structure was solved at a 2.2 Å resolution in the P212121 space group with identical unit cell constants as the single-crystal data (Supplementary Table 1).”

For me this sentence implies I will find information about the cryo-crystallographic experiment in that table. If the unit cell is identical, a rigid body refinement against cryo-data should do the trick. I think if the cryo-experiment is mentioned at all it would be interesting to add statistics & maybe do a rigid body refinement. Even in the XFEL experiments the data along the thin dimension of the crystal are quite weak, so it is reasonable to assume that cryo-data would appear anisotropic & may benefit from anisotropic truncation (especially if they were collected only on a single crystal). On a further note I may have found the explanation why molecular replacement for cryo-data failed (lattice translation disorder is probably more severe in larger crystals, see below).

“A high systematic background scattering from the direct XFEL beam (Supplementary Fig. 3) combined with pseudotranslation led to high structure refinement R-factors, although it did not affect the excellent quality of electron density maps (see Methods and Supplementary Fig. 4).”

It is indeed surprising that such an excellent electron density can be obtained with such high R-factors. When going into the structure in-depth then one makes out a couple of problems that probably result in these R-factors.

Firstly there is an issue with data completeness reported in Supplementary Table I and the real completeness (the completeness in the low resolution range drops after 6 Å down to about 80% at 30 Å).

I have no explanation for this, probably these intensities were removed as outliers in the data correction when converting to amplitudes. This could have something to do with the strange behaviour in supplementary table 4, where upon activation of the rings-grad background subtraction the low resolution multiplicity drops even below the high-resolution completeness. This is unusual and leads me to believe that there was something wrong with the low-resolution data other than the high background alone, but I am not sure since I have no experience processing data with such a high background. The authors could try to see whether the data without rings-grad despite giving worse merging statistics result in better R-factors when combined with a more conservative –push-res (see below).

When I briefly refined the structure removing the low-res data & updating solvent (I updated the solvent because I realized that there are some issues with solvent b-factors, see below) the R-factor started to drop:

In the high-res range there is likely a low real multiplicity that results in larger random errors in the reflections. Given the very high value in the –push-res option of 3.0 which would integrate crystals as follows (i believe, although it is not very clearly described in the manual of CrystFEL):

$$\text{Apparent resolution} = 5 \text{ \AA} = 0.5 \text{ nm}$$

$$1 / 0.5 \text{ nm} = 2.0 \text{ nm}^{-1}$$

$$\text{Push-res} = 3.0 \text{ nm}^{-1}$$

$$2.0 \text{ nm}^{-1} + 3.0 \text{ nm}^{-1} = 5.0 \text{ nm}^{-1}$$

$$1 / 5.0 \text{ nm}^{-1} = 0.2 \text{ nm} = 2.0 \text{ \AA}$$

So, a crystal with an apparent resolution of 5 Å (dependent on peak-detection criteria) would be integrated to 2.0 Å. Then also a crystal with an apparent resolution of 8 Å would be integrated to 2.35 Å resolution. This setting in the context of the authors data works likely like the default (to integrate everything to the detector edge). This in turn makes the high-resolution multiplicity likely inflated. It does not strongly affect the quality of the intensity data, but it will likely affect it slightly, so a more conservative push-res setting may improve statistics & make the multiplicity more reliable. It will not improve the R-factors since one reason may be incomplete convergence. In my experience low multiplicity XFEL data to high resolution can result in beautiful maps coupled to high R-factors. This has likely to do with the convergence of Monte-Carlo integration which leads to a random error on the intensities. Random intensity errors do not affect maps nearly as much as systematic ones.

Have the authors attempted to use the `--xsphere` option in order to allow for a better convergence of the data by post-refinement and the attempt to model partiality? This should clearly be attempted and the results added to Supplementary table 4, possibly together with the statistics when using a less “aggressive” `--push-res` option. It is advised to use the `--overpredict` setting in `indexamajig`, but in my experience `--xsphere` can improve low multiplicity data convergence even without `--overpredict` and can even be coupled with a slight `--push-res` of say 1.5.

As for the reported tNCS, I think there is an additional element here, which is lattice translation disorder. When looking at the model with improved R-factors and updated solvent I realized that many waters were modelled in the membrane plane. This reminded me of a case of lattice translation disorder I once came across because it results in ghost peaks from the translated chain which are close to the protein interpreted as water (it appears to happen often in LCP crystals). When overlaying chain A on top of chain B while creating a copy in order to assess low B-factors of water molecules near the sodium binding site that I thought may be potassium or chloride (see below) I saw that the resulting “layer” would also fit quite nicely with the crystal packing (this layer shift is the reason why it happens often in LCP crystals) and it may even be possible to have the shift occurring within one layer since there are almost no overlapping protein parts.

I then investigated whether there would be “ghost” Fo-Fc peaks which would fit to the translated chain and indeed I found some (Fo-Fc map at 2.5 sigma, translated chain in green):

After finding the overlap I could also see that it was along the A-axis and shifted by about 1/4th and so it was easy to identify the resulting Patterson peak (site number 2, 3rd highest peak):

Thankfully this type of defect can be readily modelled when its only along one axis, see this paper:

<https://journals.iucr.org/d/issues/2005/01/00/he5308/index.html>

If the authors need help implementing the correction, I can refer them to my colleague who did the correction for the case I mentioned. This can be quite readily implemented & should improve the R-factors depending on how strong the disorder is by 2-4% (just a guess).

Page 6

“Despite a relatively high resolution and conservation of critical sodium binding residues, such as D822.50, S1223.39, and N2987.45, we could not locate a Na⁺ in the electron density of S1P5, most likely because of a low sodium concentration in the final crystallization buffer (~20 mM). “

Has this been observed in other GPCRs? Is the affinity of this site known & have the authors considered an allosteric effect via the non-conserved residues (S81 could easily affect the affinity locally by a rotamer switch)? It appears an interesting question to answer given the relevance of sodium concentration close to the membrane in signaling and the fact that this specific sub-type of receptor is expressed preferentially in oligodendrocytes. Could a nearby water be also replaceable by potassium or chloride? Water 3007 in chain A ligated to Tyr 89 has quite a low B-factor given the surrounding residues (about half) and Water 3011 in chain B also ligated to TYR89 has an even lower one. This may point to a potassium or chloride here. However, the B-factors of the water molecules are a bit puzzling in general in this structure. While refining the solvent molecules newly, one obtains more reasonable B-factors. However, especially the waters near the sodium binding site in chain A appear to be too low, but this may be explainable by the lattice translation disorder for chain A. It may be beneficial to re-analyze these sites once the disorder was corrected for and focus on chain B.

Supplementary material:

Page 14:

Supplementary Table 1

Completeness of data in final dataset starts to be around 100% only from 6 Å onwards, why the discrepancy?

Is there a particular reason the set of free reflections is so large? I would have phenix re-assign them with the standard 2000 (although 500 are enough) and re-run a refinement until convergence after the other changes have been implemented (maybe scramble b-factors, that makes it faster).

Please give the water B-factors after re-refinement separately, they should be higher than the protein & the fact that they are lower should have pointed towards potential problems.

Why did the authors cut the resolution at 2.2 Å? The $I/\sigma(I)$, the high-res R-factor of the structure (not above ~35%) and the electron density point towards a higher resolution, the drop in CC* is probably because of the too aggressive --push-res. If in doubt do a paired refinement to increasingly higher resolution (<https://journals.iucr.org/f/issues/2021/07/00/ir5024/index.html>).

Page 17:

Supplementary Table 4

How do the authors explain the sharp drop in multiplicity in the low-resolution range when using a gradient to model the background? To me column C looks like it has the best data, did the authors try refinement against these data? Combined with a less aggressive --push-res this may be the way to go. Otherwise very nice Table and assessment of the processing progress. Great the authors included this.

Suggested improvements

Taken this analysis together I am convinced that by assessing the potential issue with the low-resolution data, improving potential high resolution multiplicity related issues, modelling the lattice translation disorder and a better refinement of the solvent (the authors may also model unexplained difference map peaks that remain after the correction and that are in the membrane plane as small lipid fragments, especially if they are close to the protein, this will help with the solvent mask determination & hence influence the R-factors of the whole structure) the authors should be able to significantly reduce the R-factors.

Nevertheless, I agree with the notion of the authors that they have a beautiful and readily interpretable electron density and I can understand why they did not seek complicated explanations when the result is so readily interpretable. However, crystallographers are ever critical especially towards serial work & a lower R-factor would help convince the "classical" crystallographer and therefore help how the paper is perceived in the community.

I suggest the authors take a closer look at the role of Tyr89 in the stabilization of the sodium binding site, since it is also in direct contact with the ligand and appears to be important.

The discussion section feels a little bit like a summary rather than a discussion, given the large amount of data presented that summary is quite important in the end, but it would be good if the authors could speculate a bit more on the role of the different sub-types of receptors and how their structural differences in the light of their data allow them to fulfill these roles.

Clarity and context

The article is clearly written and understandable and other works that are related are mentioned and compared.

References

I did not find issues with the references.

We would like to thank all Reviewers for their helpful comments and constructive critique, which we addressed below. The manuscript was revised accordingly, and all changes are highlighted in yellow. Our point-by-point responses to the Reviewers' comments are shown below in brown:

Reviewer comments

Reviewer #1

1. I had difficulties following the findings as most of them are hidden in the supporting info. I recognize there might be limits on the number of figures in the main text but the authors should try to put a figure for each part of the study. Currently, the text is disproportional to the available figures. Large sections of the text (docking, Alphafold, SNPs) are not linked to any figures. It might be good to reduce or remove these sections (see below).

We thank the reviewer for the suggestions. We modified and transferred several supporting figures into the main text as summarized below:

- 1) We combined Supplementary Figs. 7 and 8 (docking) making it a new Fig. 4 in the main text.
 - 2) We modified Supplementary Fig. 9 (SNVs) and added it to the main text as Fig. 2d.
 - 3) We modified Supplementary Figs. 10 and 11 making a new Fig. 5 for the AlphaFold section in the main text.
2. 'however, the first two residues of the P-I-F motif deviate from the consensus' – this part of the sentence should be rephrased, it gives confusion.

We appreciate the comment of the reviewer. Indeed, the sentence was unclear. Therefore, we changed it for the following:

"In all S1PRs, the dual toggle switch is conserved as L^{3.36}-W^{6.48}; however, the P-I-F motif deviates from the consensus, and in S1P₅, it is represented as I^{5.50}-V^{3.40}-F^{6.44} (Fig. 1e). Nevertheless, the I-V-F motif in S1P₅ apparently serves a similar role as the classical P-I-F motif in other receptors, as the sidechains of V^{3.40} and F^{6.44} switch over upon activation."

3. Line 219-222.

Please check these two sentences. The text contradicts Fig 2c. L292A/V has a small change in the activity compared to the wild type, whereas M296/V/W has no response. G293A has minor changes in the activity, while G293V has no response.

We thank the reviewer for catching these discrepancies and modified the corresponding sentences as follows:

"In particular, mutations L292^{7.39}A/V decrease ONO-543060 potency by over an order of magnitude, while M296^{7.43}V/W and G293^{7.40}V abolish the response to S1P (Fig. 2c). On the other hand, mutations C43^{1.39}F and G293^{7.40}A show almost no effect on ONO-543060 potency."

4. Lines 248-272:

The section of docking should be shortened or moved to the supporting info as it doesn't give much new info about the structure but further tests the position of Y89. Description of groups for ligands based on the activity without showing the chemical structures is challenging to follow. The reference to the paper doesn't help. The authors may want to give the structures of all the groups in the supporting

info. The authors give docking scores without any units, which also gives the challenge to appreciate the importance of the scoring values. Again, the SAR analysis discussion at the end of the section is difficult to grasp without the structures. Given that the authors don't validate the docking by completing mutagenesis for the selected analogs and rely mainly on the docking scores I would really recommend moving this section to the supporting info.

This part is relatively short, but we believe that it provides important insights into the conformation of Y89^{2,57} and the design of inverse agonists for S1P₅, and therefore, we prefer to keep it in the main text. For validation of our docking results we used published ligand potencies from Ref. 25. To address the Reviewer's concerns, we moved the chemical structures and their affinities from the supplementary materials to the main text (new Fig. 4) and made several additional clarifications in the text.

Finally, we added the docking score units. Although binding scores have units of energy per mole, they are not intended to accurately predict binding energies but rather provide relative ranking to distinguish good binders from poor ones.

5. Lines 273-291:

This section is very speculative and should be removed from the manuscript. It is not clear from the structure and metadynamics performed that Y89 is responsible for the inverse agonism and linked to other activation motifs such as L3.36-W6.48.

We respectfully disagree with the Reviewer and believe that our metaMD analysis of the role of Y89 in inverse agonism is valid and conclusive. Since 2021, 13 structures of S1PRs published in 5 articles^{1,2,3,4,5}, and another 3 structures will be released soon⁶. All of these structures are agonist-bound and provide a comprehensive view on the activation mechanism of the S1PR family. Meanwhile, our structural insights on inverse agonism in S1P₅ presented in this manuscript are unique and novel.

There are two independent conclusions that originated from our metaMD studies. The first one is illustrated in **Fig.3 a, c**. The reaction coordinate for this calculation is the torsion rotation of the Y^{2,57} side chain, and metaMD was done for three receptors S1P_{1,3,5}. It shows that Y^{2,57} can adopt the downward conformation only in S1P₅.

¹ Xu, Z., Ikuta, T., Kawakami, K., Kise, R., Qian, Y., Xia, R., Sun, M., Zhang, A., Guo, C., Cai, X., Huang, Z., Inoue, A. & He, Y. Structural basis of sphingosine-1-phosphate receptor 1 activation and biased agonism. *Nat. Chem. Biol.* **18**, 281–288 (2022).

² Zhao, C., Cheng, L., Wang, W., Wang, H., Luo, Y., Feng, Y., Wang, X., Fu, H., Cai, Y., Yang, S., Fu, P., Yan, W. & Shao, Z. Structural insights into sphingosine-1-phosphate recognition and ligand selectivity of S1PR3–Gi signaling complexes. *Cell Res.* **32**, 218–221 (2022).

³ Yuan, Y., Jia, G., Wu, C., Wang, W., Cheng, L., Li, Q., Li, Z., Luo, K., Yang, S., Yan, W., Su, Z. & Shao, Z. Structures of signaling complexes of lipid receptors S1PR1 and S1PR5 reveal mechanisms of activation and drug recognition. *Cell Res.* (2021). doi:10.1038/s41422-021-00566-x

⁴ Chen, H., Chen, K., Huang, W., Staudt, L. M., Cyster, J. G. & Li, X. Structure of S1PR2–heterotrimeric G13 signaling complex. *Sci. Adv.* **8**, 67 (2022).

⁵ Maeda, S., Shiimura, Y., Asada, H., Hirata, K., Luo, F., Nango, E., Tanaka, N., Toyomoto, M., Inoue, A., Aoki, J., Iwata, S. & Hagiwara, M. Endogenous agonist-bound S1PR3 structure reveals determinants of G protein–subtype bias. *Sci. Adv.* **7**, 1–12 (2021).

⁶ Yu, L., He, L., Gan, B., Ti, R., Xiao, Q., Hu, H. & Zhu, L. Structural Insights into Sphingosine-1-phosphate Receptor Activation. (2022). *bioRxiv*. doi:10.1101/2022.01.15.475352

The second conclusion is illustrated in **Fig.3 b, d**. The reaction coordinate here is the torsion rotation of L^{3.36}, and metaMD was done for S1P₅ with either upward or downward orientation of Y^{2.57}. It shows that the active conformation of the L^{3.36} microswitch is incompatible with the downward orientation of Y^{2.57}, which can explain the mechanism of inverse agonism in S1P₅.

To better describe the metaMD method we replaced lines 544-545 as follows:

“We employed a metadynamics (metaMD) approach⁴¹ to estimate free energy profiles along the rotation of the χ_1 torsion angle in the side chain of Y^{2.57} in S1P₁, S1P₃, and S1P₅ as well as free energy profiles along the rotation of the χ_1 torsion angle in the side chain of L^{3.36} in S1P₅ with two alternative orientations of Y^{2.57}. This method is based on addition of biasing repulsive ...”

In line 549, after “The metaMD simulations were run for 10 ns each.” we inserted:

“Two conformations corresponding to the free energy minima along the rotation of the χ_1 torsion of Y^{2.57} in S1P₅ were selected for the subsequent metaMD simulations of L^{3.36}, in which the orientation of Y^{2.57} was harmonically restrained in the upward or downward positions.”

We also added the following sentence in the results section, line 276:

“We used metaMD to estimate free energy profiles along the reaction coordinate corresponding to the torsion rotation of the L119^{3.36} side chain for S1P₅ with Y89^{2.57} restrained in the upward and downward orientations. The upward orientation ...”.

6. Line 292-321:

Supporting Figure 9 should be in the main text.

We agree with the Reviewer and added an improved Supplementary fig. 9 to the main text (now Fig. 2d).

7. Line 322-363:

the authors should comment on whether any structures of the S1PR family were in the training set of AlphaFold. The results of VS should be in the main text.

The VS results were already included in the main text (the last paragraph of the section “Comparison with AlphaFold predictions”). In this revision, we added two panels (Fig. 5g-h) to illustrate them.

No structures of the S1PR family were included in the AlphaFold training set. We clarified this both in the main text and in the methods.

Reviewer #2

1. The authors indicate that the rotamer conformation of Y89^{2.57} is correlated with the activation state of the receptor, and this aromatic residue serves as a selectivity determinant for S1P₅-specific ligands. In the known structure of agonist siponimod-bound S1P₅ receptor, Y89^{2.57} assumes an upward orientation, but it does not make direct contact with the agonist (Yuan et al., Cell Research, 2021). According to the functional assay conducted in this study, the Y89A or Y89W mutations show similar effects on the potencies of S1P agonism and ONO-5430608 antagonism. Could the authors provide an explanation for these observations?

Although Y89 does not make direct contacts with the agonist siponimod, this residue occupies a very important place in the receptor. First of all, its conformation can directly modify the shape of the binding pocket or it can indirectly influence ligand response by changing conformations of adjacent residues. Secondly, Y89 is located between TM1 and TM7 on the putative entrance pathway for lipid-like ligands⁷, and, thirdly, it directly affects the activation state of the toggle switch L^{3.36}-W^{6.48}. Therefore, mutations of Y89 can have complex and not readily interpretable consequences on receptor signaling. The lack of significant effects of the Y89W mutation on ONO-5430608 may reflect the fact that a moderate increase in the side chain volume can be well accommodated in the inactive state of the receptor. On the other hand, in the active state, when this residue is oriented upward, Y89W may indirectly affect other residues in the pocket, slightly reducing the S1P potency. The Y89A mutant is however not tolerated and becomes irresponsive to S1P, and, therefore, ONO-5430608 antagonism could not be measured.

2. In the structure of ONO-54306-bound S1P5 receptor, Y89^{2.57} forms direct contact with the tetrahydro-benzazepine double ring of ONO-5430608. Could the downward conformation of Y89^{2.57} be induced by the inverse agonist ONO-5430608?

Furthermore, all the ligands in Fig. S8 have a core double ring but different substituents that correspond to different receptor-binding affinities. Could the core double ring system determine inverse agonism of these ligands by forcing a downward orientation of Y89^{2.57}?

Yes, we agree with the Reviewer. The downward conformation of Y89^{2.57} is likely induced or stabilized by the inverse agonist ONO-5430608 and its analogs. The tetrahydro-benzazepine double ring indeed plays an important role forming contacts with Y89^{2.57} in the downward orientation. However, substituents of the benzene ring are also important since they clash with the upward Y89^{2.57} orientation.

We based our conclusions from molecular docking on published SAR data for tetrahydronaphthalene derivatives⁸. It is likely that both the core double ring system and the substituents determine inverse agonism, however, we could not confirm or reject it at the current stage.

3. The authors indicated that the mutational effects of residues lining the allosteric subpocket are correlated with their occluded area while interacting with ONO-543060. The residue I93^{2.61} has a large occluded area (127 Å²) when ONO-543060 is bound to the receptor, similar to L292^{7.39} and M296^{7.43}. Why wasn't this residue tested in functional assays?

Although I93^{2.61} has a large occluded surface area, it is located further away from the ligand compared to other mutated residues. The closest distance between I93^{2.61} and ligand atoms is 4.3 Å, while it is 3.7 Å and 3.6 Å for L292^{7.39} and M296^{7.43}, respectively. For this reason I93^{2.61} was not selected for the mutational studies.

4. Some naturally occurring mutations such as C43F in the putative ligand entrance gateway, show little effect in functional tests. Could the authors speculate what might be the underlying reason for this?

⁷ Hanson, M. A., Roth, C. B., Jo, E., Griffith, M. T., Scott, F. L., Reinhart, G., Desale, H., Clemons, B., Cahalan, S. M., Schuerer, S. C., Sanna, M. G., Han, G. W., Kuhn, P., Rosen, H. & Stevens, R. C. Crystal structure of a lipid G protein-coupled receptor. *Science* **335**, 851–855 (2012).

⁸ Watanabe, T., Kusumi, K. & Yuichil Inagaki, I. (2017). Tetrahydronaphthalene Derivative. United States patent No. US 2019/0031605A1 U.S. Patent and Trademark Office. <https://patentscope.wipo.int/search/en/detail.jsf?docId=US236791513>

The amino acid residues in the ligand entrance gateway should only affect the kinetics of the ligand binding but not the equilibrium constants, including functional cAMP response that was measured in our assays.

5. Fig. S2a shows that the purified S1P₅ receptor protein is still quite heterogeneous based on results from analytical size exclusion chromatography. Why wasn't size exclusion chromatography applied on preparative scale to further purify the protein?

We typically do not apply preparative size exclusion chromatography (SEC) when we purify GPCR samples for crystallization in LCP, because LCP itself filters out residual aggregates during protein reconstitution. An additional SEC step leads to at least a 20% drop in the final protein yield without an improvement in crystallization outcome.

6. The authors indicated that the EC₅₀ value of ONO-5430608 in Gi-mediated cAMP accumulation assay is 1.7nM (page 6 under the section 'overall structure of the ligand-binding pocket'). This is different from the data reported in Table S2 (pEC₅₀=8.54M). Please clarify the difference. Was this value derived experimentally or cited from literature?

We thank the Reviewer for catching this mistake. We corrected the value for pEC₅₀ in Supplementary Table 2 to 8.77, which is the same as EC₅₀=1.7 nM reported in the main text. We also updated other pEC₅₀ values in Supplementary Table 2 as they were calculated without including the "zero concentration" point. All corrections are relatively small and does not affect any conclusions in this study. This update does not affect Fig. 2c as this figure already had correct pEC₅₀ values. We apologize for this oversight.

Figures:

1. The residues in Fig. 3a are not correctly labeled. The S1P1 receptor should have M124^{3,32} and V301^{7,43}. The S1P5 receptor has V115^{3,32} and M296^{7,43}.

We thank the Reviewer for pointing out these discrepancies. We corrected the residue labels in the revised Fig. 3a.

2. The authors may consider adding a panel in Fig. 1 that shows only the new structure described in this manuscript.

We thank the Reviewer for the suggestion, however, we doubt that adding a panel in already crowded Fig. 1 would be useful. Nevertheless, we added an additional panel d to Fig. 2 with mapping important SNVs on the overall S1P₅ structure.

Reviewer #3

Page 4:

1. "Attempts to solve the structure were first made using cryocooled crystals at a synchrotron source achieving a maximum of 4 Å single-crystal data resolution, however, the obtained data could not be phased using molecular replacement."

The reader does not know anything about this experiment and how seriously it was attempted, given a crystal size of 30 μm it is clear that the dose should have been spread over multiple crystals, which is standard for GPCR cryo-crystallography at synchrotrons. The passage creates the impression that a

much higher resolution was obtained at the XFEL without doing an experiment that would show this, is therefore misleading and should be removed.

We agree with this comment and replaced the above statement with the following:

“Our initial attempts at solving the structure using synchrotron data were unsuccessful.”

2. “The crystal structure was solved at a 2.2 Å resolution in the P212121 space group with identical unit cell constants as the single-crystal data (Supplementary Table 1). “

For me this sentence implies I will find information about the cryo-crystallographic experiment in that table. If the unit cell is identical, a rigid body refinement against cryo-data should do the trick. I think if the cryo-experiment is mentioned at all it would be interesting to add statistics & maybe do a rigid body refinement. Even in the XFEL experiments the data along the thin dimension of the crystal are quite weak, so it is reasonable to assume that cryo-data would appear anisotropic & may benefit from anisotropic truncation (especially if they were collected only on a single crystal). On a further note I may have found the explanation why molecular replacement for cryo-data failed (lattice translation disorder is probably more severe in larger crystals, see below).

The data that we collected at a synchrotron were very weak and of poor quality. Therefore, we decided against including their analysis in this manuscript and replaced the above statement with the following:

“The crystal structure was solved at a 2.2 Å resolution in the P212121 space group (Supplementary Table 1). “

3. “A high systematic background scattering from the direct XFEL beam (Supplementary Fig. 3) combined with pseudotranslation led to high structure refinement R-factors, although it did not affect the excellent quality of electron density maps (see Methods and Supplementary Fig. 4).”

It is indeed surprising that such an excellent electron density can be obtained with such high R-factors. When going into the structure in-depth then one makes out a couple of problems that probably result in these R-factors.

Firstly there is an issue with data completeness reported in Supplementary Table I and the real completeness (the completeness in the low resolution range drops after 6 Å down to about 80% at 30 Å).

I have no explanation for this, probably these intensities were removed as outliers in the data correction when converting to amplitudes. This could have something to do with the strange behaviour in supplementary table 4, where upon activation of the rings-grad background subtraction the low resolution multiplicity drops even below the high-resolution completeness. This is unusual and leads me to believe that there was something wrong with the low-resolution data other than the high background alone, but I am not sure since I have no experience processing data with such a high background. The authors could try to see whether the data without rings-grad despite giving worse merging statistics result in better R-factors when combined with a more conservative –push-res (see below).

Indeed, the low resolution intensities were removed by the French-Wilson correction (FW) in Phenix. However, it is not related to the rings-grad background subtraction but is a result of some computational singularity that appeared when FW is applied to data at their whole resolution range after CrystFEL (90-1.6 Å). The completeness of the data is about 100% for all resolutions when FW is applied to the refinement resolution range (30.0-2.2 Å). We used these 100% complete data to re-refine the model in this revision.

We also tried using data without rings-grad as proposed (column C' in the revised Supplementary Table 4) but it did not make any improvement in Rfree of the model (see Table 1 below and explanation therein for more details).

When I briefly refined the structure removing the low-res data & updating solvent (I updated the solvent because I realized that there are some issues with solvent b-factors, see below) the R-factor started to drop:

We believe that the R-factor improvement observed by the Reviewer was mainly related to the water update rather than the low resolution removal. We compared two refinement jobs with water update using data in 6.0-2.2 Å and 30.0-2.2 Å range. Rfree dropped by ~1% in both cases.

The Rfree improvement is probably related to masking phantom lattice translocation disorder (LTD) densities by water molecules as was also mentioned by the reviewer. As we describe below, we corrected our data for LTD and updated water molecules of our model in this revision.

In the high-res range there is likely a low real multiplicity that results in larger random errors in the reflections. Given the very high value in the `-push-res` option of 3.0 which would integrate crystals as follows (i believe, although it is not very clearly described in the manual of CrystFEL):

Apparent resolution = 5 Å = 0.5 nm

$1 / 0.5 \text{ nm} = 2.0 \text{ nm}^{-1}$

Push-res = 3.0 nm⁻¹

$2.0 \text{ nm}^{-1} + 3.0 \text{ nm}^{-1} = 5.0 \text{ nm}^{-1}$

$1 / 5.0 \text{ nm}^{-1} = 0.2 \text{ nm} = 2.0 \text{ Å}$

So, a crystal with an apparent resolution of 5 Å (dependent on peak-detection criteria) would be integrated to 2.0 Å. Then also a crystal with an apparent resolution of 8 Å would be integrated to 2.35 Å resolution. This setting in the context of the authors data works likely like the default (to integrate everything to the detector edge). This in turn makes the high-resolution multiplicity likely inflated. It does not strongly affect the quality of the intensity data, but it will likely affect it slightly, so a more conservative push-res setting may improve statistics & make the multiplicity more reliable. It will not improve the R-factors since one reason may be incomplete convergence. In my experience low multiplicity XFEL data to high resolution can result in beautiful maps coupled to high R-factors. This has likely to do with the convergence of Monte-Carlo integration which leads to a random error on the intensities. Random intensity errors do not affect maps nearly as much as systematic ones.

Have the authors attempted to use the `-xsphere` option in order to allow for a better convergence of the data by post-refinement and the attempt to model partiality? This should clearly be attempted and the results added to Supplementary table 4, possibly together with the statistics when using a less

“aggressive” --push-res option. It is advised to use the --overpredict setting in indexamajig, but in my experience --xsphere can improve low multiplicity data convergence even without --overpredict and can even be coupled with a slight --push-res of say 1.5.

We thank the Reviewer for this suggestion. Unfortunately, applying less aggressive push-res values didn't help to increase the overall data quality. Following the Reviewer's advice, we re-integrated data with the --overpredict option, and tried to merge data (with/without overpredict) using different push-res values from 1.0 to 3.0 and different partiality models (unity/xsphere). Columns H and I (revised Supplementary Table 4) represent best (as judged by the CC* value in the “high-resolution” shell) merging statistics without and with --overpredict, respectively. Unfortunately, they do not show improvements compared to the data obtained earlier, column G.

New merging strategies were added to the revised Supplementary Table 4. To choose the best of them, we refined the structure against data I, H, F and the original dataset - G (see Table 1 below and explanation therein for more details). Rfree shows no improvement compared to G. Taking into account that the G dataset has slightly better overall and outer shell merging statistics (see Rsplit and I/sigma), we used it for the final refinement.

As for the reported tNCS, I think there is an additional element here, which is lattice translation disorder. When looking at the model with improved R-factors and updated solvent I realized that many waters were modelled in the membrane plane. This reminded me of a case of lattice translation disorder I once came across because it results in ghost peaks from the translated chain which are close to the protein interpreted as water (it appears to happen often in LCP crystals). When overlaying chain A on top of chain B while creating a copy in order to assess low B-factors of water molecules near the sodium binding site that I thought may be potassium or chloride (see below) I saw that the resulting “layer” would also fit quite nicely with the crystal packing (this layer shift is the reason why it happens often in LCP crystals) and it may even be possible to have the shift occurring within one layer since there are almost no overlapping protein parts.

I then investigated whether there would be “ghost” Fo-Fc peaks which would fit to the translated chain and indeed I found some (Fo-Fc map at 2.5 sigma, translated chain in green):

After finding the overlap I could also see that it was along the A-axis and shifted by about 1/4th and so it was easy to identify the resulting Patterson peak (site number 2, 3rd highest peak):

```

There are      6 peaks higher than the threshold      0.49244 (  3.00000 *sigma)

These peaks are sorted into descending order of height, the top  6 are selected for output
The number of symmetry related peaks rejected for being too close to the map edge is  0
Peaks related by symmetry are assigned the same site number

Order No. Site Height/Rms   Grid      Fractional coordinates   Orthogonal coordinates
  1   1   1  205.94     0   0   0  0.0000  0.0000  0.0000     0.00  0.00  0.00
  2   3   3   77.10    32  72  0  0.3755  0.5000  0.0000    22.47  51.71  0.00
  3   2   2   16.97    21   0  0  0.2489  0.0000  0.0000    14.89   0.00  0.00
  4   4   4   14.10     0   0  7  0.0000  0.0000  0.0279     0.00  0.00  5.24
  5   5   5    6.55    31  72  7  0.3736  0.5000  0.0272    22.35  51.71  5.12
  6   6   0    4.07    42   0 128  0.5000  0.0000  0.5000    29.92   0.00  93.95

```

Thankfully this type of defect can be readily modelled when its only along one axis, see this paper: <https://journals.iucr.org/d/issues/2005/01/00/he5308/index.html>

If the authors need help implementing the correction, I can refer them to my colleague who did the correction for the case I mentioned. This can be quite readily implemented & should improve the R-factors depending on how strong the disorder is by 2-4% (just a guess).

Indeed, we noticed the presence of a pseudotranslation-related modulation in the diffraction intensities before (described in lines 102 and 518-519 of the original manuscript). However, we did not assign it to the LTD and are thankful to the reviewer for raising this issue and particularly for the recommendations about the data treatment.

Besides the ghost densities and the Patterson peak mentioned by the reviewer, we noticed that some synchrotron diffraction patterns contain diffuse reflections interspersed with sharp reflections. These three features are symptoms of LTD as described in the recent review⁹.

Following the reviewer's suggestion, we corrected reflections according to Wang et al.¹⁰ as well as used an alternative approach described by Ponnusamu et al.¹¹ We applied both approaches to the original dataset used in the manuscript for the refinement (dataset G in the revised Supplementary table 4) and datasets I, H and C' that resulted from alternative merging approaches proposed by the reviewer 3.

Coordinates and B-factors were randomized prior to the refinement for each of the dataset to reduce possible bias using phenix.refine options sites.shake=0.3 adp.randomize=true. Improvements in the refinement R-factors are shown below in Table 1.

	R/Rfree (%)
--	-------------

⁹ Lovelace, J. J., & Borgstahl, G. E. (2020). Characterizing pathological imperfections in macromolecular crystals: lattice disorders and modulations. *Crystallography reviews*, 26(1), 3-50.

¹⁰ Wang, J., Kamtekar, S., Berman, A. J., & Steitz, T. A. (2005). Correction of X-ray intensities from single crystals containing lattice-translocation defects. *Acta Crystallographica Section D: Biological Crystallography*, 61(1), 67-74.

¹¹ Ponnusamy, R., Lebedev, A. A., Pahlow, S., & Lohkamp, B. (2014). Crystal structure of human CRMP-4: correction of intensities for lattice-translocation disorder. *Acta Crystallographica Section D: Biological Crystallography*, 70(6), 1680-1694.

	G (original dataset)	C'	H	I
no LTD correction	30.57/33.50	33.06/35.41	31.73/34.13	31.37/34.41
Wang correction ¹⁰	29.98/32.86	32.31/35.16	30.44/33.63	30.55/33.90
Ponnusamy correction ¹¹	29.85/32.91	31.47/34.25	30.55/33.36	30.80/33.35

Table 1. Improvement of model R-factors after LTD-correction.

The lowest Rfree was archived using the G dataset with the Wang correction. This dataset was used for refining the revised model.

We updated section “Structure determination and refinement” in the Methods (lines 519 - 521 of the revised manuscript) to describe the applied LTD correction:

“The relatively high Rfree of the structure can be partially explained by the high systematic background scattering and the modulation in the diffraction intensities. The modulation is produced by two factors: (1) the NCS operator $(x, y, z) \rightarrow (1/8 + x, -y, -z)$ seen as a Patterson peak at $(3/8, 1/2, 0)$ with a 0.3 of the origin peak height, and (2) the lattice-translocation defect (LTD)⁶² seen as a Patterson peak at $(1/4, 0, 0)$ with a 0.1 of the origin peak height. We corrected our data partially for LTD as described previously⁶³ that resulted in an Rfree drop by 0.6% during the refinement.”

Page 6

“Despite a relatively high resolution and conservation of critical sodium binding residues, such as D82^{2.50}, S122^{3.39}, and N298^{7.45}, we could not locate a Na⁺ in the electron density of S1P₅, most likely because of a low sodium concentration in the final crystallization buffer (~20 mM). “

Has this been observed in other GPCRs? Is the affinity of this site known & have the authors considered an allosteric effect via the non-conserved residues (S81 could easily affect the affinity locally by a rotamer switch)? It appears an interesting question to answer given the relevance of sodium concentration close to the membrane in signaling and the fact that this specific sub-type of receptor is expressed preferentially in oligodendrocytes.

Yes, this has been observed in other GPCRs. Despite the high conservation of the sodium binding site in class A GPCRs, a sodium ion was modeled only in 8 out of 96 unique class A GPCR structures, according to GPCRdb. Since sodium ion is relatively light and does not produce anomalous scattering, its detection relies on the ability to resolve coordination and distances to coordinating protein and solvent atoms, which typically requires a better than 2.5 Å resolution. Therefore the most common reason a sodium ion is not observed in GPCR structures is their relatively low resolution. An interesting example is the dopamine receptor D4, for which high-resolution structures were determined both with and without a sodium ion (Wang et al, 2017, Science 358, 341). Importantly, an electron density for sodium was observed only when it was added during crystallization at a concentration of 200 mM, about 2 times higher than its binding affinity ~85 mM. The sodium binding affinity was measured for several GPCRs, ranging between few and few hundred mM (see for review Zarzycka et al 2019, Pharmacol Rev 71, 571), however, it is not known for S1P₅. S81^{2.49} belongs to the canonical sodium binding site in Class A GPCRs where it is conserved as A (58%), S (17%), or T (17%). The structure of the prostaglandine D2 receptor

(PDB ID 7M8W) contains S^{2.49} which interacts with sodium and has the same conformation as S81^{2.49} in our S1P5 structure.

Taken together, we believe that S1P₅ should bind sodium ion (as most class A GPCRs), but we did not observe it in our structure most likely due to the fact that the concentration of Na⁺ in the final crystallization buffer was lower than its apparent affinity.

Could a nearby water be also replaceable by potassium or chloride? Water 3007 in chain A ligated to Tyr 89 has quite a low B-factor given the surrounding residues (about half) and Water 3011 in chain B also ligated to TYR89 has an even lower one. This may point to a potassium or chloride here. However, the B-factors of the water molecules are a bit puzzling in general in this structure. While refining the solvent molecules newly, one obtains more reasonable B-factors. However, especially the waters near the sodium binding site in chain A appear to be too low, but this may be explainable by the lattice translation disorder for chain A. It may be beneficial to re-analyze these sites once the disorder was corrected for and focus on chain B.

Potassium or chloride ions have never been observed in GPCR structures in this location. We re-analysed water molecules using revised data. After LTD correction, the difference between the mean water and protein B-factors became smaller (11.4 Å² before to 3.7 Å² after LTD correction), although the mean B-factor is still slightly higher for protein (44.5 Å² for protein vs 40.8 Å² for water molecules). This is likely related to our inability to fully correct LTD in SFX data. Extremely low B-factors (<10 Å²) appear randomly for waters in both chains and for all data merging strategies shown in Table 1. Notably, waters with low B-factors are different for different refinement protocols and datasets. Therefore, we cannot rely on water B-factors for ion placement.

Supplementary material:

Page 14:

Supplementary Table 1

Completeness of data in final dataset starts to be around 100% only from 6 Å onwards, why the discrepancy?

The completeness reported in Supplementary Table 1 was for the dataset before applying the FW correction. As we have mentioned above, some of the low resolution intensities were rejected by the FW correction in Phenix. This problem is addressed in the revised data where completeness is about 100% in the whole resolution range.

Is there a particular reason the set of free reflections is so large? I would have phenix re-assign them with the standard 2000 (although 500 are enough) and re-run a refinement until convergence after the other changes have been implemented (maybe scramble b-factors, that makes it faster).

The free set was assigned automatically. We re-assigned free reflections (the total number is 1999 now), randomized all B-factors, annealed the model and refined until convergence taking into account other reviewer's suggestions (see the full list of changes in the answer to "Suggested improvements").

Please give the water B-factors after re-refinement separately, they should be higher than the protein & the fact that they are lower should have pointed towards potential problems.

Following the suggestions of the Reviewer, we updated water molecules and re-refined the structure after correction for LTD. We included the mean water B-factor in the Supplementary Table 1. It is still

slightly lower than the mean protein B-factor (44.5 \AA^2 for protein vs 40.8 \AA^2 for water molecules), which is likely related to our inability to properly correct for LTD in SFX data.

Why did the authors cut the resolution at 2.2 \AA ? The $I/\sigma(I)$, the high-res R-factor of the structure (not above $\sim 35\%$) and the electron density point towards a higher resolution, the drop in CC^* is probably because of the too aggressive --push-res. If in doubt do a paired refinement to increasingly higher resolution (<https://journals.iucr.org/f/issues/2021/07/00/ir5024/index.html>).

The resolution cut off was determined by paired refinement and remained the same for the revised model. We added an explanation to the Method section (line 515 of the revised manuscript):

“The final resolution cutoff was determined by paired refinement⁶⁴.”

Page 17:

Supplementary Table 4

How do the authors explain the sharp drop in multiplicity in the low-resolution range when using a gradient to model the background? To me column C looks like it has the best data, did the authors try refinement against these data? Combined with a less aggressive --push-res this may be the way to go. Otherwise very nice Table and assessment of the processing progress. Great the authors included this.

As we mentioned above, some of the low resolution intensities were rejected by the French-Wilson correction in Phenix. This problem is addressed in the revised data where completeness is about 100% in the whole resolution range. We also tried using data without rings-grad and with a less aggressive push-res as proposed (column C' in the revised Supplementary Table 4) but it did not result in Rfree improvement (see Table 1 above and explanation therein for more details).

Suggested improvements

Taken this analysis together I am convinced that by assessing the potential issue with the low-resolution data, improving potential high resolution multiplicity related issues, modelling the lattice translation disorder and a better refinement of the solvent (the authors may also model unexplained difference map peaks that remain after the correction and that are in the membrane plane as small lipid fragments, especially if they are close to the protein, this will help with the solvent mask determination & hence influence the R-factors of the whole structure) the authors should be able to significantly reduce the R-factors.

Nevertheless, I agree with the notion of the authors that they have a beautiful and readily interpretable electron density and I can understand why they did not seek complicated explanations when the result is so readily interpretable. However, crystallographers are ever critical especially towards serial work & a lower R-factor would help convince the “classical” crystallographer and therefore help how the paper is perceived in the community.

We are thankful to the Reviewer for very valuable comments. Following these suggestions we:

- solved the issue with the low completeness of the low-resolution data,
- assessed various proposed merging strategies,
- modeled LTD,
- re-assigned the free set to 1,999 reflections,
- randomized coordinates and B-factors, performed simulated annealing,
- updated solvent molecules and refined the structure to convergence.

As a result, the model Rfree was reduced by 2.8% from 35.7 to 32.9%.

I suggest the authors take a closer look at the role of Tyr89 in the stabilization of the sodium binding site, since it is also in direct contact with the ligand and appears to be important.

In this manuscript, we extensively analyzed the role of Tyr89 in inverse agonism of S1P₅. This residue, however, does not belong to the canonical sodium binding site, nor could we locate sodium in the electron density. Therefore, we do not have any experimental data to further interrogate this question, although this residue does coordinate the network of water molecules inside the receptor.

The discussion section feels a little bit like a summary rather than a discussion, given the large amount of data presented that summary is quite important in the end, but it would be good if the authors could speculate a bit more on the role of the different sub-types of receptors and how their structural differences in the light of their data allow them to fulfill these roles.

We appreciate this suggestion and agree that it would be an interesting topic for a follow up review article. Such analysis, however, is clearly outside the scope of this study, which is focused on inverse agonism in S1P₅, and will be distracting from the main message presented here.

REVIEWERS' COMMENTS

Reviewer #1 (Remarks to the Author):

The authors have addressed and clarified all my comments. I recommend publishing this manuscript.

Reviewer #2 (Remarks to the Author):

The authors have addressed all of my concerns.

Reviewer #3 (Remarks to the Author):

The points I suggested have been fully addressed and the manuscript is ready for publication.